# Fluid extraction from the left-right organizer uncovers mechanical properties needed for symmetry breaking

Pedro Sampaio[1†], Sara Pestana[1†], Catarina Bota[1], Adán Guerrero[2], Ivo A Telley[3], David Smith[4,5]*, Susana Santos Lopes[1]*

[1]CEDOC, Chronic Diseases Research Centre, NOVA Medical School. Faculdade de8 Ciências Médicas, Universidade Nova de Lisboa, Lisboa, Portugal; [2]Laboratorio Nacional de Microscopía Avanzada. Departamento de Genética del Desarrollo y Fisiología Molecular. Instituto de Biotecnología. Universidad Nacional Autónoma de México (UNAM), Cuernavaca, Mexico; [3]Instituto Gulbenkian de Ciência, Fundação Calouste Gulbenkian, Oeiras, Portugal; [4]School of Mathematics, University of Birmingham, Birmingham, United Kingdom; [5]Institute for Metabolism and Systems Research, University of Birmingham, Birmingham, United Kingdom

**\*For correspondence:**
d.j.smith@bham.ac.uk (DS);
susana.lopes@nms.unl.pt
(SSantosL)

[†]These authors contributed equally to this work

**Competing interest:** The authors declare that no competing interests exist.

**Abstract** Humans and other vertebrates define body axis left-right asymmetry in the early stages of embryo development. The mechanism behind left-right establishment is not fully understood. Symmetry breaking occurs in a dedicated organ called the left-right organizer (LRO) and involves motile cilia generating fluid-flow therein. However, it has been a matter of debate whether the process of symmetry breaking relies on a chemosensory or a mechanosensory mechanism (Shinohara et al., 2012). Novel tailored manipulations for LRO fluid extraction in living zebrafish embryos allowed us to pinpoint a physiological developmental period for breaking left-right symmetry during development. The shortest critical time-window was narrowed to one hour and characterized by a mild counterclockwise flow. The experimental challenge consisted in emptying the LRO of its fluid, abrogating simultaneously flow force and chemical determinants. Our findings revealed an unprecedented recovery capacity of the embryo to re-fil and re-circulate new LRO fluid. The embryos that later developed laterality problems were found to be those that had lower anterior angular velocity and thus less anterior-posterior heterogeneity. Next, aiming to test the presence of any secreted determinant, we replaced the extracted LRO fluid by a physiological buffer. Despite some transitory flow homogenization, laterality defects were absent unless viscosity was altered, demonstrating that symmetry breaking does not depend on the nature of the fluid content but is rather sensitive to fluid mechanics. Altogether, we conclude that the zebrafish LRO is more sensitive to fluid dynamics for symmetry breaking.

## Editor's evaluation

Sampaio and colleagues manipulate fluid dynamics in zebrafish Kupffer's vesicle to ask if fluid movement or something in the fluid governs the break in symmetry. These results support a role for fluid movement and detection as important in breaking symmetry in a ciliated left-right organizer and help set a time window when fluid flow is critical for this process.

## Introduction

Vertebrate organisms break lateral symmetry during embryonic development and define left and right. The left-right axis is the third and final body axis to be established in the embryo and encompasses a biophysical process yet to be fully elucidated. In mouse, frogs, and fish it involves fluid movement in

a specialized organ called the left right organizer (LRO). In the mouse embryo, cilia driven fluid flow inside the LRO has been proposed to activate either a mechanosensory mechanism in the epithelial cells that constitute the LRO, or to transport morphogens (as dissolved molecules or carried inside extracellular vesicles) towards the left side of the LRO (*Nonaka et al., 1998*; *McGrath et al., 2003*; *Tanaka et al., 2005*). While the mechanism is unknown, it culminates in asymmetric left-sided calcium signalling at the node cells and a subsequent conserved Nodal pathway on the left side of the embryo (*Tanaka et al., 2005*; *Takao et al., 2013*; *Mizuno et al., 2020*). The endpoint of this developmental process is the asymmetric localization of internal visceral organs, such as the heart and liver.

Analysis of cilia-driven LRO flow is therefore crucial to understand this initial process of symmetry breaking. The transparent zebrafish embryo is a particularly suitable model system enabling flow visualization and subsequent scoring of organ asymmetry in the same individual. We and others have shown that wildtype (WT) zebrafish embryos have a stereotyped flow pattern in the LRO (known as Kupffer's vesicle or KV), exhibiting higher dorsal anterior speed (*Essner et al., 2005*; *Kramer-Zucker et al., 2005*; *Supatto et al., 2008*; *Sampaio et al., 2014*; *Smith et al., 2014*; *Ferreira et al., 2017*). Our group has demonstrated that deviations from this flow pattern due to less cilia or shorter motile cilia, predictably generate larvae with wrong laterality outcomes, indicating a biological relevance to flow (*Sampaio et al., 2014*). Thus, flow dynamics and the mechanical-chemical readout are central to solving the problem of how left-right asymmetry is first established.

Although the LRO is an organ with a limited lifetime, from ~2 ss to 14 ss (*Oteíza et al., 2008*), seminal work unveiled that the zebrafish Kupffer's vesicle is dispensable for left-right establishment after 10 somite stages (ss) (*Essner et al., 2005*). Guided by this finding, most subsequent studies have been performed at 8–10 ss, when an asymmetry in *dand5* gene expression is first detected (*Lopes et al., 2010*; *Sampaio et al., 2014*; *Ferreira et al., 2017*; *Ferreira et al., 2019*). The decrease of *dand5* gene expression on the left side of the LRO is the first asymmetric molecular signal in left-right development, downstream of fluid flow, an event conserved across species (*Marques et al., 2004*; *Hojo et al., 2007*; *Schweickert et al., 2010*).

Yuan and colleagues have shown that, as soon as zebrafish LRO cilia are formed and some start beating, at around 1–4 ss, intraciliary calcium oscillations (ICOs) can be registered in LRO cells (*Yuan et al., 2015*). These signals were reported to precede cytosolic calcium elevations that extend mostly to the left mesendodermal regions and were shown to be essential to trigger correct left-right patterning. Therefore, it is conceivable that an asymmetry in *dand5* mRNA expression, first detectable at 8 ss in zebrafish (*Lopes et al., 2010*), results from molecular events that significantly precede that time point. Although it is evident from pioneer studies using mouse mutants without cilia or with immotile cilia that fluid flow has a critical role in conveying a LR biased signal to the nearby cells (*Nonaka et al., 1998*; *Supp et al., 1999*), both the nature of the signal and its detection mechanism remain elusive. The mechanosensory hypothesis postulates that fluid flow can be mechanically sensed by cilia, through a Polycystin protein complex involving PKD1L1 and PKD2 cation channel (*Field et al., 2011*; *Kamura et al., 2011*; *Yuan et al., 2015*). Conversely, the chemosensory hypothesis posits that vesicular parcels or extracellular vesicles containing an unknown chemical determinant are transported by the directional fluid flow towards the left side of the mouse node (*Cartwright et al., 2020*). However, the existence of such signaling morphogen has not yet been confirmed or any evidence for extra-cellular vesicle endocytosis or content release has ever been revealed with the exception of the work from Hirokawa's lab (*Tanaka et al., 2005*).

The difficulty in testing which hypothesis is correct relies on the fact that it is virtually impossible to perform genetic manipulations of the LRO cell secretion pathway without impacting cilia length or motility. This is not surprising if we think that any perturbation in lipid or protein trafficking affects motile cilia function, which reflects an experimental impossibility, because to test chemosensation we need to keep fluid mechanics constant. To address these 20 year-long-standing questions, we decided to take a step back and unequivocally determine the developmental period for left-right initiation. We envisaged that by determining the critical time-window within the 7 hr lifetime of the zebrafish LRO, we could then characterize its fluid flow and, thus, narrow down the variables that contribute to left-right establishment. Ultimately, we aimed at obtaining evidence for one or the other model for LR initiation as current scenarios fail to fully uncouple the LRO hydrodynamics from the potential role of fluid flow in transporting vesicles or molecules with signalling properties (*Cartwright et al., 2020*). The LRO is filled with fluid, but the nature of the fluid content is unknown because, so far,

its minute volume, around one hundred picolitres (*Roxo-Rosa et al., 2015*), prevented a detailed biochemical analysis. However, the swelling mechanism of the LRO lumen is known to be driven by the chloride channel known as cystic fibrosis transmembrane conductance regulator (CFTR) that, when impaired, predictably leads to empty LROs and left-right defects (*Navis et al., 2013*). Thus, it is possible to block the swelling process by using pharmacology, anti-sense technology or mutants for CFTR, without affecting cilia number or motility. However, manipulation with precision and higher resolution in time was lacking.

To determine the sensitive time-window for LR initiation, we devised a novel assay that allows for highly controlled mechanical manipulation of the LRO fluid. We performed total fluid extractions from the LRO at different developmental time points only 30 min apart. After confirming that the KV was still present and able to refill, we followed the development of each embryo until organ laterality could be determined. This approach led us to narrow the time-window for symmetry breaking to 1-hr, from 4 to 6 ss, corresponding to the cell rearrangement period mechanistically explained by *Wang et al., 2011* and initially noted by *Kreiling et al., 2007* and *Okabe et al., 2008*. During this time interval, we found fluid flow to be mild but already directional. Extraction of LRO fluid at 5 ss led to 35% of WT embryos with both heart and liver incorrect localization. Embryos did not develop LR defects when the original KV fluid was replaced with Danieau's buffer, but LR axis was perturbed when viscosity of the same buffer was increased. Altogether, our experiments suggest that fluid flow mechanics, contrary to the nature of fluid content, has a major role in LR initial establishment.

## Results

### Fluid extraction affects LR during 1-hr time-window

KV fluid flow was disrupted by performing fluid extraction at each LRO developmental stage using a microscope-based micromanipulation setup connected to microfluidic actuators (*Figure 1A*). We envisaged that if we found the sensitive time-period, we could then investigate with more precision the properties that were necessary and sufficient to break LR. For this purpose, we challenged WT embryos by extracting the liquid from their LRO lumen from 3 to 12 ss (see extraction *Figure 1—video 1* at 8 ss and *Figure 1—video 2* at 5 ss). Manipulated embryos that recovered the LRO liquid, were later characterized according to heart and liver laterality. This procedure intendedly depleted the LRO fluid transiently, disrupting the flow generated by motile cilia, while at the same time it potentially extracted any molecular signals that are present in the fluid.

Controls for these experiments included confirmation that cilia number, length and anterior-posterior (A/P) distribution were not perturbed after fluid extraction at 5 ss, in fixed samples at 8 ss (*Figure 1—figure supplement 1*) and that number of motile / immotile cilia and their A/P distribution was not changed when evaluated by live imaging at 6 ss (*Figure 1—figure supplement 2* and *Supplementary file 1*). Concomitantly, we analysed cilia beat frequency (CBF) of motile cilia, by high-speed video microscopy. Sham controls, where the LRO was perforated with the extraction needle without applying suction, were included. At 6 ss, no significant difference was found between Sham controls with an average of 38.93 Hz (n=56 cilia; 6 embryos) and fluid extracted embryos with 37.85 Hz (n=60 cilia; 6 embryos); (*Figure 1—figure supplement 2*).

To determine if after extraction at 5 ss there were significant anterior-posterior cilia distribution differences between 'Sham' and manipulated experimental groups the spatial distribution of cilium types was compared by live imaging as well as the ratio of motile to immotile cilia (*Figure 1—video 3*). Consistent with previous studies (*Tavares et al., 2017*), cilia from control 'Sham' embryos at 6 ss (n=7) were predominantly motile (82±4 SEM %) and localized preferentially to the anterior half of the LRO (Fisher test, *P*-value <0.05, *Figure 1—figure supplement 2*). The total number of motile cilia for each analysed 'Sham' embryo revealed some intrinsic embryo variability even among the control group as reported before (*Sampaio et al., 2014*). Similarly, in the manipulated group (n=6 embryos), motile cilia were more abundant (78 ± 10 %) and were distributed more anteriorly (Fisher test p-value <0.05; *Figure 1—figure supplement 2*). Therefore, no significant differences were found for the ratio of motile / immotile cilia between manipulated group compared to 'Sham' controls (*Figure 1—figure supplement 2*).

Although most embryos recovered the fluid extraction procedure without LR defects (*Figure 1B*), some presented both heart and liver misplacements, starting with intervention at 3 ss (2/13 embryos,

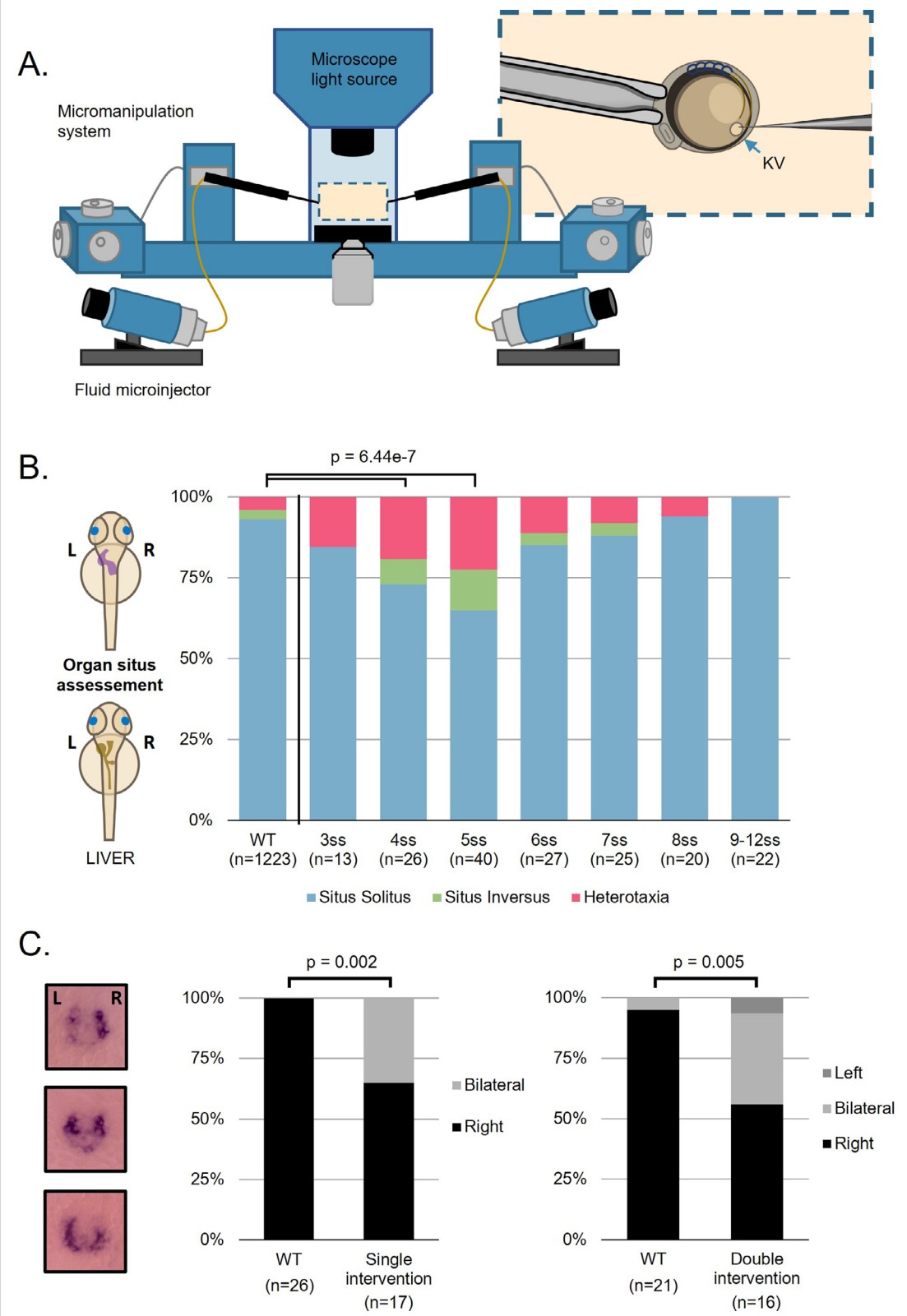

**Figure 1.** Fluid extractions uncover a sensitive one-hour interval. (**A**) Schematic representation on the micromanipulation setup developed for the zebrafish LRO fluid extraction throughout development. (**B**) Evaluation of organ patterning after 48 hr. Heart and liver position were scored for each embryo manipulated in the respective developing stage from 3 to 12 ss, *situs inversus* in zebrafish means both organs are localized on the right side

*Figure 1 continued on next page*

*Figure 1 continued*

and heterotaxia means that at least one organ is wrongly localized. (**C**) *dand5* expression pattern at 8 ss after single intervention at 5 ss and double intervention at 5 and 7 ss for LRO liquid extraction. LRO. left-right organizer, ss. somite stage.

The online version of this article includes the following video, source data, and figure supplement(s) for figure 1:

**Source data 1.** Characterization of heart and gut situs in zebrafish larvae.

**Figure supplement 1.** Manipulated embryos develop normal LRO cilia evaluated by immunofluorescence.

**Figure supplement 1—source data 1.** Cilia number for WT and manipulated embryos.

**Figure supplement 2.** Cilia Beat Frequency and motile/ immotile cilia ratio do not change in manipulated embryos evaluated by live imaging.

**Figure supplement 2—source data 1.** Live imaging cilia motility status and localization.

**Figure supplement 3.** LRO areas from recovering embryos that develop left-right defects are not different from embryos that have a normal development.

**Figure supplement 3—source data 1.** Left right organizer area quantification.

**Figure 1—video 1.** Close-up of fluid extraction technique employed in a 8 ss embryo.
https://elifesciences.org/articles/83861/figures#fig1video1

**Figure 1—video 2.** Close-up of fluid extraction technique employed in a 5 ss embryo.
https://elifesciences.org/articles/83861/figures#fig1video2

**Figure 1—video 3.** Evaluation of motile and immotile cilia distribution at 6 somite stage embryos.
https://elifesciences.org/articles/83861/figures#fig1video3

15%, p-value = 0.235), becoming more pronounced for intervention at 4 ss (7/26 embryos, 27%, Fisher test, p-value = 0.002) and peaking with intervention at 5 ss (14/40 embryos, 35%; Fisher test, p-value = 6.44e-7). LR organ misplacements then became progressively non-significant from 6 ss onwards (4/27 embryos, 19%; Fisher test, p-value = 0.865; *Figure 1B*). At this point, as a response to the fluid extraction challenge, we had identified the most sensitive time-window for LR establishment. Next, we wanted to investigate in detail what were the factors that did not allow some of the challenged embryos to recover the correct laterality.

To elucidate how fluid depletion affected LR signalling, we examined the expression pattern of *dand5*. It is well established that in WT embryos, zebrafish *dand5* is mainly expressed on the right side of the LRO at 8 ss (*Lopes et al., 2010*) due to left-sided decrease of *dand5* mRNA. Thus, we first evaluated *dand5* expression pattern at 8 ss in embryos that were manipulated at 5 ss, which was the developmental stage with highest incidence of LR defects upon liquid extraction. We observed that *dand5* expression also became abnormally bilateral in 35% of the cases (6/17 manipulated embryos compared to 0/26 controls, Fisher test, p-value = 0.002; *Figure 1C*) suggesting that either lack of fluid flow or depletion of a molecular signal prevented left-sided degradation of *dand5* mRNA. Next, to further challenge the system we performed an additional fluid extraction 1 hr after the first intervention when enough liquid was replenished, at 7 ss. Results showed a similar proportion of bilateral *dand5* expression (6/16, 37.5%) to that observed at 5 ss (comparing both plots in *Figure 1C*, Fisher test p-value = 0.728). This experiment confirmed that fluid extractions at 7 ss no longer affect LR development significantly, as already implied in *Figure 1B*. To continue the investigation on a potential deficiency in flow recovery in the 'LR Defects' group after fluid extraction at 5 ss, we next assessed how fluid flow dynamics varied with time by retrospectively comparing embryos with and without LR defects.

## LRO recovers lumen area after fluid extraction

The lumen of manipulated LROs started to expand soon after fluid extraction (*Figure 1—figure supplement 3*), indicating that the fluid secretion machinery mediated by CFTR (*Navis et al., 2013*; *Roxo-Rosa et al., 2015*) was not collaterally affected by the manipulation procedure and was able to drive the swelling of the LRO lumen. Our observations contrasted with previous reports (*Essner et al., 2005*; *Hojo et al., 2007*) of irreversibly emptied LROs after manipulations, highlighting that the technical approach developed in this work confers minimal damage to the LRO core structure.

*Gokey et al., 2016* showed that below 1300 $\mu m^2$ of KV cross-sectional area at 8 ss, LR patterning could be compromised. Therefore, a scenario where some LROs replenish better than others could explain a differential recovery, enabling the correct organ patterning in most, but not all embryos.

Thus, we first investigated if there were significant differences in fluid recovery between manipulated embryos that developed LR defects from those that did not. We performed LRO fluid extraction at 5 ss and followed the recovery of the LRO cross-sectional area and fluid dynamics along development (at 6 ss, 7 ss and 8 ss, *Figure 1—figure supplement 3A*). As expected, the LROs from the control group ('Sham') were larger at each sampled time point (t-test, p-value <0.05) compared to the fluid extracted groups Further, our results showed that the LRO area was similar as well as the area change per somite (t-test, pvalue >0.05) in manipulated embryos that developed a normal organ patterning ('No Defects' group) and those that developed incorrect LR axis establishment 'Defects' group, (*Figure 1—figure supplement 3B, C*).

## Fluid flow angular velocity is reduced in manipulated embryos showing LR developmental defects

Flow can be determined by following native particles in KV as a proxy for tracking fluid directly. The general pattern of fluid flow in the LRO is roughly circular with local variations (*Sampaio et al., 2014*; see *Figure 2—video 1* for an embryo that developed normal LR development and *Figure 2—video 2* for an embryo that showed abnormal LR development). Local flow speed is a useful measure for identifying regional flow patterns, but it does not fully describe the directional material transport of the LRO fluid (*Ferreira et al., 2017*; *Juan et al., 2018*). Thus, we determined the angular velocity at discrete locations as a proxy for the circular flow speed (*Figure 2A–C*) and show angular velocity components: tangential and radial velocities in *Figure 2—figure supplements 1 and 2*, respectively. Considering the two sensing hypotheses, we predicted that any type of detectable LR signal, either as flow-induced shear force or as a signalling molecule, must happen in the vicinity of the cell membrane. Taking this into consideration, we analysed the fluid dynamics within the outer half of the radius of the LRO lumen.

Angular velocity data from the many tracks obtained from each of the sham and extraction experiments were analysed at 6, 7, and 8 ss, first dividing the LRO in 8 radial sections and plotting the median angular velocity per section (*Figure 2A–C*). To account for the hierarchical structure of the data with both within- and between-embryo variability in angular velocity tracks, linear mixed effects regression was employed to characterise how angular velocity varied across the LRO and over time (*Figure 2D–E*), incorporating somite stage, normalized LR/ AP axis position and group (Sham/ Defects/ No Defects) as independent variables. Coefficient values can be interpreted as showing relative effect sizes and direction (see *Table 1*).

The linear mixed regression for experiments with fluid extraction and 'Sham' intervention at 5 ss identified the following significant fixed (embryo-independent) effects: (i) a significant reduction of angular velocity in the 'Defects group' (p-value = 1.2e-5, coefficient –0.13 rad/s), (ii) greater angular velocity in the anterior compared with posterior region of the LRO (p-value = 6.3e-98, coefficient 0.16 rad/s), (iii) increased flow velocity over time (p-value = 7.1e-6, coefficient 0.021 rad/s/stage), (iv) enhanced left-right difference in the 'No Defects group' (p-value = 0.006, coefficient 0.027 rad/s), (v) reduced posterior-anterior difference in the 'Defects group' (p-value = 9.5e-5, coefficient –0.041 rad/s), (vi) enhanced posterior-anterior difference in the 'No Defects group' (p-value = 0.002, coefficient 0.034 rad/s), (vii) enhanced degree of velocity increase over time in the 'No Defects' group (p-value = 2.3e-5, coefficient 0.027 rad/s/stage) and (viii) (at marginal significance level) some degree of velocity increase over time in the 'Defects group' (p-value = 0.041, coefficient 0.013 rad/s/stage).

It is important to note that the increase in anterior angular velocity (*Figure 2D*) in the 'No defects' group compared with the 'Sham group' can be explained by the change in size of the Kupffer's vesicle after extraction (*Figure 1—figure supplement 3B*): assuming that cilia beating and, therefore, material transport at the periphery is the same in 'Sham' group and 'No-defect' group, the fluid flow absolute velocity is also unchanged, but in a smaller sphere (R = ~25 µm versus R = ~30 µm) it leads to a proportionally higher angular velocity. In the 'Defects' group, despite this geometric effect, the angular velocity suggests that flow velocity was decreased.

Next, to clarify for further differences we showed that radial velocity was modestly stronger and positive on the left-anterior sections of the LRO (*Figure 2—figure supplement 2*) denoting directionality towards the membrane was enhanced in the 'No defects' group.

Further, between the 'Defects' group and the other groups we investigated the directionality of the individual particles as it could provide more cues for localized differences that we observed denoted

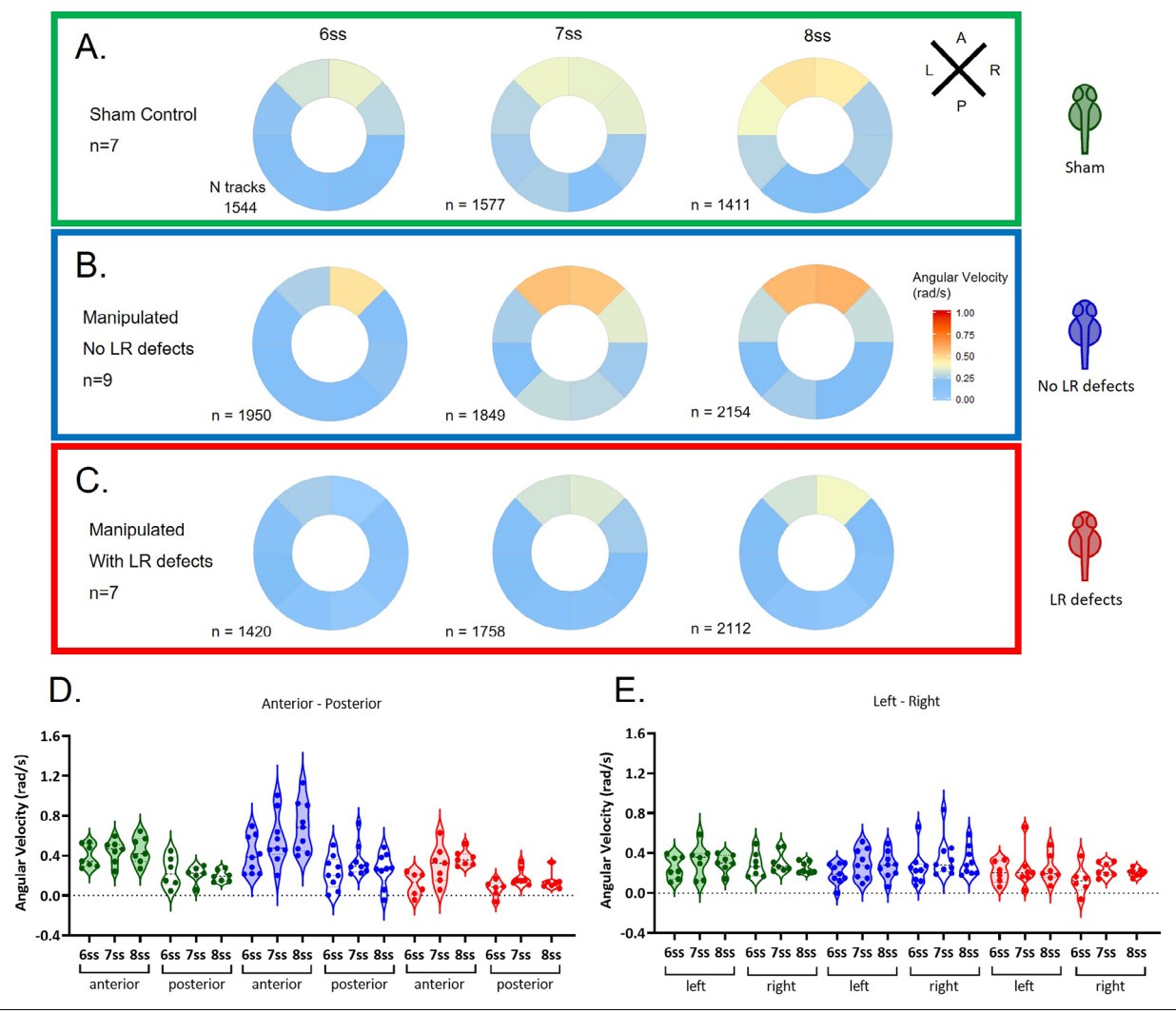

**Figure 2.** Angular velocity during LRO fluid flow recovery. Angular velocity polar plots for 6 ss, 7 ss, and 8 ss for the three different groups (**A**) 'Sham' control embryos colour coded in green, (**B**) 'No LR Defects' group of embryos colour coded in blue and (**C**) 'LR Defects' group of embryos colour coded in red. LR defects refer to misplacement of heart or liver encountered at the larval stage after fluid extraction was performed at 5 ss; number of tracks refers to the number of particle trajectories identified for the quantifications and respective angular velocity plots. Colour code on polar plots refers to the median angular velocity for pooled embryos. (**D–E**) Violin plots showing quantifications of angular velocities found for the tracks analysed (**D**) anterior-posterior and (**E**) left and right. Dots contained in the violin plots correspond to median values per embryo. A statistical linear mixed effects regression was applied (see results in *Table 1*).

The online version of this article includes the following video, source data, and figure supplement(s) for figure 2:

**Source data 1.** Angular velocity quantifications at 6, 7, and 8 somite stages after fluid extraction at 5 somite stage.

**Figure supplement 1.** Tangential velocity component of particle movement in the outer LRO luminal space.

**Figure supplement 2.** Radial velocity components of particle movement in the outer KV luminal space during fluid recovery.

**Figure 2—video 1.** Example of a manipulated embryo that developed normal zebrafish *situs* (left heart and left liver).
https://elifesciences.org/articles/83861/figures#fig2video1

**Figure 2—video 2.** Example of a manipulated embryo that developed abnormal *situs*.
https://elifesciences.org/articles/83861/figures#fig2video2

by negative values for angular velocity indicating the existence of some particles with clockwise flow. We divided the LROs into sections of 30 degrees each and by looking at the direction of each tracked particle in detail (*Figure 3A* and *Figure 3—figure supplements 1–6*), we only found significant differences in directionality when comparing the group 'Defects' with the 'Sham' control group in anterior

**Table 1.** Linear mixed-effects regression: Results for fluid extraction experiment.

Fixed effects coefficient estimates are given. Number of observations: 15775; embryos n=23; Fixed effects coefficients: 12; Random effects coefficients 69; Covariance parameters: 7. Regression formula: AAV ~1 + Group*LR +Group*PA +Group*Time + (1+Group | EmbryoID). Results relate to *Figure 2D–E*; *** indicates p<0.001; ** indicates p<0.01; * indicates p<0.05.

| Name | Estimate | Lower CI | Upper CI | p-value |
|---|---|---|---|---|
| (Regression line intercept) | 0.2515 | 0.20762 | 0.29544 | 3.8408E-29 *** |
| No defects group | -0.032775 | -0.12321 | 0.05766 | 0.47748 |
| Defects group | -0.12728 | -0.18434 | -0.070215 | 1.2396E-05 *** |
| Left-right axis | 0.0042934 | -0.0092686 | 0.019363 | 0.57655 |
| Posterior-anterior axis | 0.16131 | 0.14636 | 0.17626 | 6.285E-98 *** |
| Somite stage | 0.02145 | 0.012091 | 0.030809 | 7.0847E-06 *** |
| Interaction of no defects group and left-right axis | 0.026653 | 0.007577 | 0.04573 | 0.006176 ** |
| Interaction of defects group and left-right axis | -0.0026553 | -0.022753 | 0.017443 | 0.79567 |
| Interaction of no defects group and posterior-anterior axis | 0.033645 | 0.012118 | 0.055171 | 0.0021912 ** |
| Interaction of defects group and posterior-anterior axis | -0.041406 | -0.06219 | -0.020622 | 9.461E-05 *** |
| Interaction of no defects group and somite stage | 0.026635 | 0.014317 | 0.038953 | 2.2637E-05 *** |
| Interaction of defects group and somite stage | 0.013497 | 0.00056621 | 0.026427 | 0.040778 * |

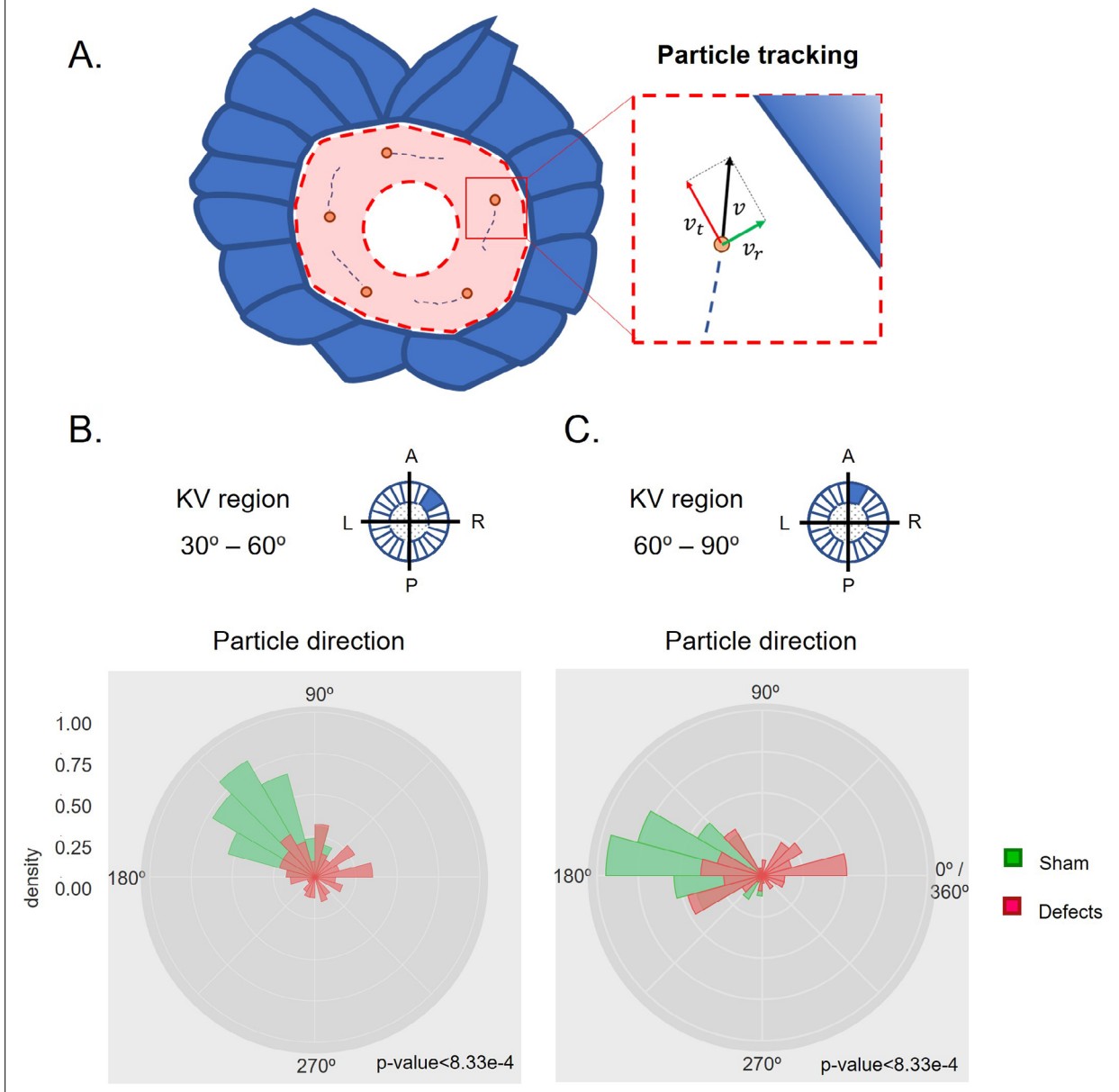

**Figure 3.** Directionality of vector fields changes in embryos that develop LR defects. (**A**) Diagram representing particle directionality. LRO area sections were delimited based on intervals of 30 degrees. Highlighted are regions (**B**) from 30 to 60 degrees and (**C**) 60–90 degrees, that showed significant differences in particle movement between the groups 'Sham' control embryos and embryos with 'LR Defects'. Each density plot represents the pooled tracked trajectory of all moving particles at any given point in time. Respective area region analysed is represented on the top right corner of each plot. To assess differences upon fluid manipulation 'Sham' and 'Defect' groups were plotted for the 6 ss. Kolmogrov-Smirnov test was used for comparing trajectory distribution between the two groups. Full data can be found in *Figure 3—figure supplements 1–6*.

The online version of this article includes the following figure supplement(s) for figure 3:

**Figure supplement 1.** Particle movement direction at different regions of the LRO for 'Sham' and 'No Defects' groups at 6 ss.

**Figure supplement 2.** Particle movement direction at different regions of the LRO for 'Sham' and 'No Defects' groups at 7 ss.

**Figure supplement 3.** Particle movement direction at different regions of the LRO for 'Sham' and 'No Defects' groups at 8ss.

**Figure supplement 4.** Particle movement direction at different regions of the KV for 'Sham' and 'Defects' groups at 6 ss.

**Figure supplement 5.** Particle movement direction at different regions of the LRO for 'Sham' and 'Defects' groups at 7 ss.

**Figure supplement 6.** Particle movement direction at different regions of the LRO for 'Sham' and 'Defects' groups at 8 ss.

sectors of the LRO at 6 ss (*Figure 3B and C*). In this region the 'Defects' group presented disoriented particle trajectories compared to both the 'Sham' control group and the 'No Defects' group at 6 ss. This suggests that the manipulation might have regionally perturbed the fluid so that at 6 ss, the earlier time point after extraction, the effect of the transient manipulation was still detectable. Conversely, the 'No Defects' group never showed significant differences in directionality compared to the 'Sham' control group (*Figure 3—figure supplement 1*; see also *Figure 3—figure supplements 2–6*) for complete data on all sectors of the LRO for 7–8 ss.

In summary, our data showed that after fluid extraction at 5 ss, the group of embryos that later developed LR defects showed a decreased angular velocity, preferentially at the anterior LRO region resulting in reduced AP difference and it also presented deviations from the counterclockwise flow direction anteriorly.

The study of the angular velocity highlighted that a robust flow recovery mechanism is in place in most zebrafish WT embryos, ensuring that LR succeeds in most of the manipulated embryos. Both the groups of embryos that recovered or failed to recover LR development after the extraction manipulation, exposed new sensitive mechanical properties of the anterior region of the LRO that will be discussed below.

### A mechanosensory mechanism likely drives LR asymmetry

Extraction of the LRO fluid annihilated both flow dynamics and fluid content, thus it did not allow to discern a potential mechanosensory mechanism from a chemosensory one. Benefitting from the micromanipulation setup we next designed an experiment to help us to uncouple these two sensory systems. We therefore diluted the fluid content in a physiological buffer (Danieau buffer, DB) without

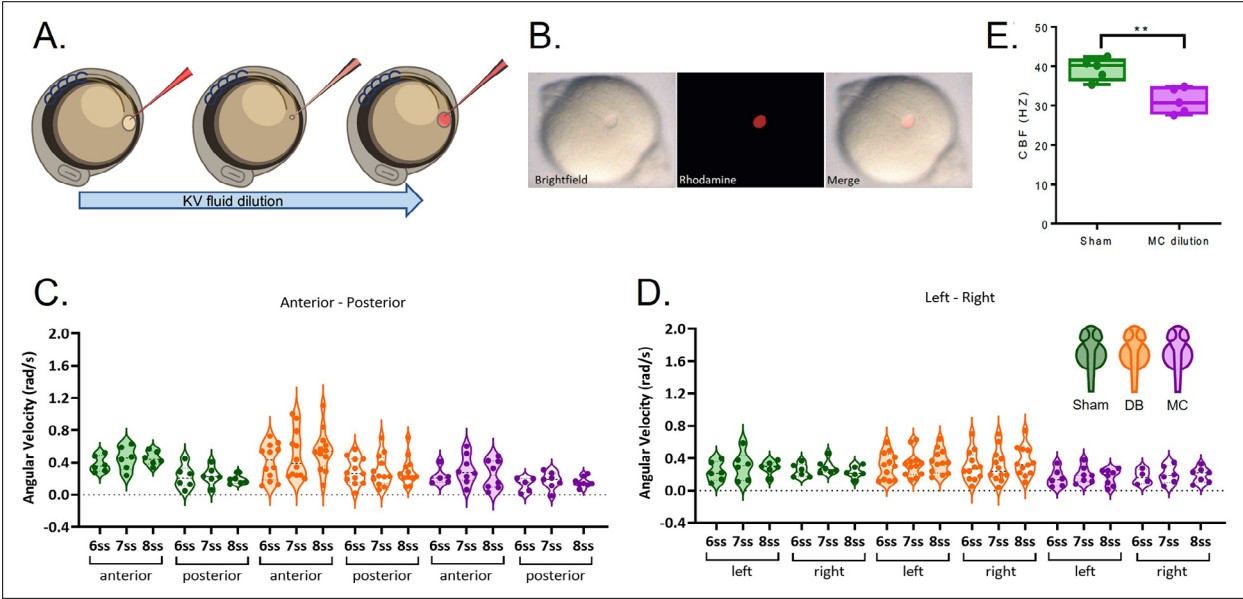

**Figure 4.** LRO fluid dilution with physiological buffer has no negative impact on angular velocity. (**A**) Diagram of the dilution experiment: KV fluid is extracted by a needle previously loaded with fluorescent rhodamine-dextran diluted in Danieau's buffer (DB) that, after mixing of the two liquids, are re-injected into the LRO. (**B**) Example of a successfully micro-injected embryo labelled with fluorescent rhodamine-dextran. (**C–D**) Violin plots showing quantifications of angular velocities found for the tracks analysed (**C**) anterior-posterior and (**D**) left and right. Dots contained in the violin plots correspond to median values per embryo. A statistical linear mixed effects regression was applied (see results in *Table 2*).

The online version of this article includes the following video, source data, and figure supplement(s) for figure 4:

**Source data 1.** Cilia beat frequency quantification during methylcellulose treatment.

**Source data 2.** Angular velocity quantifications at 6, 7 and 8 somite stages after fluid extraction at 5 somite stage and replacement with normal or viscous Danieau buffer.

**Figure supplement 1.** LRO flow dynamics is only affected when fluid dilution alters fluid viscosity (**A**) Angular velocity polar plots for 6 ss, 7 ss and 8 ss for the three different groups (**A**) 'Sham' control, (**B**) 'DB dilution' group and (**C**) 'MC dilution' group.

**Figure 4—video 1.** Close-up of fluid dilution technique employed in a 5 ss embryo.

https://elifesciences.org/articles/83861/figures#fig4video1

**Table 2.** Linear mixed-effects regression: Results for Danieau's buffer (DB) and methylcellulose (MC) dilution experiments.

Fixed effects coefficient estimates are given. Number of observations: 16622; n=30 embryos; Fixed effects coefficients: 12; Random effects coefficients 90; Covariance parameters: 7. Results relate to *Figure 4C–D*. Regression formula: AAV ~1 + Intervention*LR +Intervention*PA +Intervention*Time + (1+Intervention | EmbryoID). *** indicates p<0.001; ** indicates p<0.01; * indicates p<0.05.

| Name | Estimate | Lower CI | Upper CI | p-value |
|---|---|---|---|---|
| (Regression line intercept) | 0.25148 | 0.20891 | 0.29405 | 6.8705E-31 *** |
| DB injection | 0.070724 | –0.030824 | 0.17227 | 0.17223 |
| MC injection | –0.12344 | –0.19256 | –0.05431 | 0.00046626 *** |
| Left-right axis | 0.0042626 | –0.008177 | 0.016702 | 0.50182 |
| Posterior-anterior axis | 0.16135 | 0.14901 | 0.17369 | 4.4137E-142 *** |
| Somite stage | 0.021413 | 0.013687 | 0.029139 | 5.6349E-08 *** |
| Interaction: DB injection and left-right axis | –0.012356 | –0.027636 | 0.002925 | 0.11301 |
| Interaction: MC injection and left-right axis | 0.026366 | 0.008478 | 0.044254 | 0.0038681 *** |
| Interaction: DB injection and posterior-anterior axis | –0.02139 | –0.036997 | –0.00578 | 0.0072282 *** |
| Interaction: MC injection and posterior-anterior axis | –0.073208 | –0.092273 | –0.05414 | 5.4857E-14 *** |
| Interaction: DB injection and time | –0.012635 | –0.022248 | –0.00302 | 0.0099937 *** |
| Interaction: MC injection and time | –0.0004407 | –0.012174 | 0.011293 | 0.94131 |

changing the flow dynamics for more than a few seconds, so that we could test whether the fluid content was crucial for LR establishment. To address how this dilution experiment affected flow dynamics, we monitored angular velocity over time as before, from 6 to 8 ss. As a positive control for flow dynamics perturbation, we used methylcellulose (MC, viscosity ~1500 cP; water, viscosity 1 cP) as previously used in *Xenopus* and mouse (*Schweickert et al., 2007*; *Shinohara et al., 2012*) to make the fluid more viscous and cause LR defects.

First, we extracted the LRO fluid from 5 ss embryos and diluted it in approximately 1 µl volume of Danieau buffer (DB) previously loaded into the needle. Next, after a few seconds, we re-injected the resulting mixed liquid (*Figure 4A*). To test if dilution of the fluid content was indeed occurring, we performed the same experiment using a rhodamine tracer and confirmed that the LRO lumen became fluorescent red after the procedure (*Figure 4B*). The data for median angular velocities is presented in *Figure 4—figure supplement 1* and the application of a linear mixed effects regression for this set of experiments with DB and MC dilution versus 'Sham' controls at 5 ss identified the following significant fixed effects (*Figure 4C–D*; *Table 2*): (i) as expected we saw a significant reduction of angular velocity following MC dilution (p-value = 4.7e-4, coefficient –0.12 rad/s), but not DB alone; (p-value = 0.17), (ii) as noted before controls showed greater velocity in the anterior compared to the posterior region of the LRO (p-value = 4.4e-142, coefficient 0.16 rad/s), (iii) controls also showed increased flow velocity over time (p-value = 5.6e-8, coefficient 0.021 rad/s/stage), (iv) an unpredicted faster flow on the right compared with the left following MC dilution was highlighted (p-value = 0.0039, coefficient 0.026 rad/s), but not by DB dilution alone (p-value = 0.11) (v) we noted a modest difference in flow between anterior and posterior following DB dilution (p-value = 0.0072, coefficient –0.021 rad/s), (vi) very weak difference in flow between anterior and posterior following MC dilution (p-value = 5.5e-14, coefficient –0.073 rad/s), which has a negative effect on LR outcome, and (vii) there is some evidence for an effect of DB dilution (e.g. in the anterior) receding over time (p-value = 0.010, coefficient –0.013 rad/s/stage).

Sham control embryos (in green), embryos with LRO fluid diluted with DB (in orange) and embryos diluted with methylcellulose, MC, (in purple). (E) CBF of the motile beating cilia in the 'Sham' control embryos versus embryos with LRO fluid diluted with MC. KV: Kupffer's vesicle; ss: somite stage; DB Danieau's buffer; MC methylcellulose.

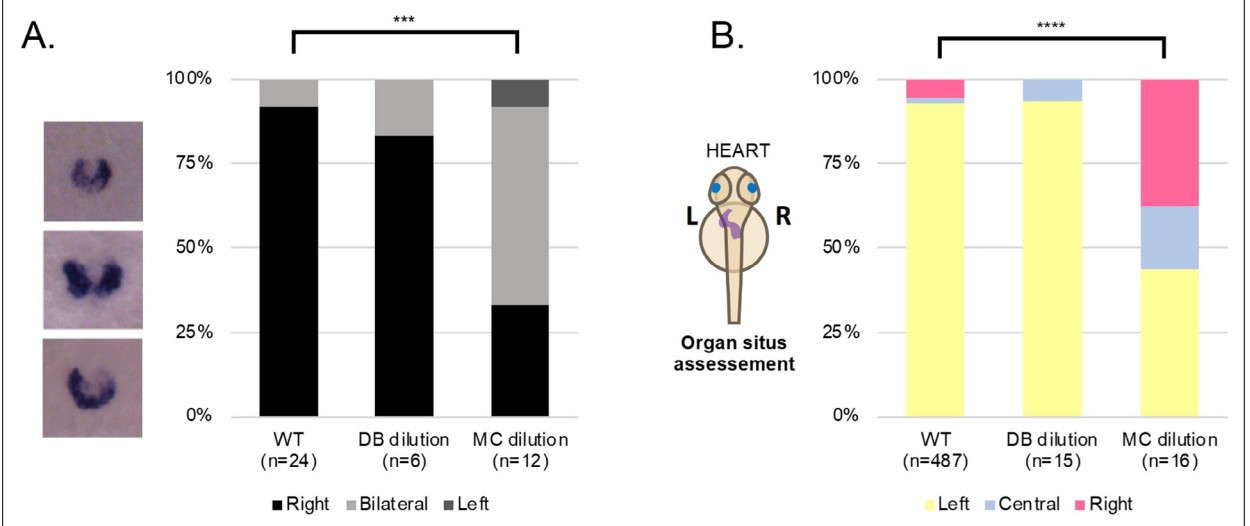

**Figure 5.** LRO fluid content does not affect left-right development. (**A**) *dand5* mRNA expression pattern in wild-type control embryos, embryos with LRO fluid diluted with DB and embryos with LRO fluid diluted with MC. Embryos were manipulated at 5 ss and fixed at 8 ss for *dand5* whole mount embryo in situ hybridization. (**B**) Heart position (left, central or right sided) was scored at 30 hpf in wild-type control embryos, embryos with LRO fluid diluted with DB and embryos with LRO fluid diluted with MC. ss: somite stage; DB Danieau buffer; MC methylcellulose; hpf: hours post fertilization.

The online version of this article includes the following source data for figure 5:

**Source data 1.** Quantification of *dand5* expression pattern and heart situs.

So importantly, the DB dilution experiment showed that quickly replacing the LRO fluid by a physiological buffer (DB) has no negative impact on angular velocity while the positive control (MC) has a great negative effect, as expected. We also observed that in the positive control, dilution of LRO fluid with MC significantly reduced cilia beat frequency (31.17 Hz ±1.4; n=50 cilia, 5 embryos) as compared to 'Sham' controls (39.29 Hz ±1.2 n=55 cilia; 5 embryos; *Figure 4E*, Student's t-test with Welch's correction, p-value <0.01).

Moreover, these experiments demonstrated that following addition of MC but not DB alone, the resulting fluid flow affected *dand5* expression pattern, with more embryos displaying bilateral expression (7/12 embryos, 58%; *Figure 5A* Fisher test, p-value <0.001). Consistent results were obtained later in development for organ placement, with 56% of manipulated embryos showing misplacement of the heart position after MC treatment but not DB dilution alone (*Figure 5B*; Fisher test, p-value <0.0001). Overall, these experiments provide evidence that fluid content is not as relevant as fluid flow mechanics for symmetry breaking. Indeed, if a signalling molecule contained in the LRO fluid was needed for *dand5* mRNA degradation, then the prediction would be that the DB dilution experiment resulted in many more embryos with *dand5* bilateral expression pattern than it did, as well as more heart misplacements.

Our observations suggest that symmetry breaking is insensitive to the LRO fluid composition. Alternatively, if we attempt to interpret these results considering the chemosensory hypothesis, dilution of extracellular vesicles (EVs) with signalling molecules within, or secreted morphogens could only occur without impacting LR if its secretion rate and activity were sufficiently high and fast, respectively, to overcome the strong diffusion effect of the LRO flow dynamics, a matter that will be discussed below.

## A mathematical model of flow disruption predicts the experimental data

Our experimental data showed that one hour time-window, from 4 to 6 ss, is the most sensitive period to the fluid extraction challenge, exposing a relevant shorter developmental timing for early LR establishment. In addition, our data also demonstrated that anterior angular velocity is a crucial flow property. To better interpret the intervention results further and elucidate the probable importance of each somite stage in left-right patterning, we constructed a probabilistic mathematical model of how anterior angular velocity, integrated from 3 ss to 8 ss, produces a symmetry-breaking signal.

Anterior angular velocity was modelled as a random variable, with mean and variance changing in time, and in response to intervention, based on mixed effects regression on observed data and previously reported observations of cilia density.

The unknown model parameters were the sensitivity weightings of the LRO to this angular velocity signal at each stage, with normal development occurring provided the integrated weighted signal exceeding a given threshold. Model parameters were then the six weightings relative to this threshold; parameters were fitted to observed results by simulating 1000 virtual LROs for each intervention. Details of the model construction and fitting procedure are given in Supplemental data and *Figure 6—figure supplements 1 and 2*.

The best fit for the full model showed good correspondence with observed experimental data (*Figure 1B* and *Figure 6A*) in particular, the highest defect rate close to 40% following intervention at 5 ss, and double intervention (5 ss +7 ss) producing a similar defect rate. This model assigned almost all of the stage weighting to 3 ss - 6 ss (*Figure 6B*). Taking account both stage weighting and anterior angular velocity to quantify the contribution of each stage, nearly 90% of the contribution to symmetry breaking occurred in the 4 ss – 6 ss interval (*Figure 6C*; note that tall bars indicate a high contribution relative to other stages).

To assess if models in which only later stages contributing to symmetry breaking could also explain the results, reduced models with each of 3 ss, 3–4 ss, 3–5 ss and 3–6 ss set to zero sensitivity were constructed in a similar manner to assess if observed results could in principle be matched without the contribution of these early stages (*Figure 6D–O*). The model omitting 3 ss contributions matched observed data similarly to the full model (*Figure 1B* and *Figure 6D*). Omitting both 3 ss and 4 ss contributions led to qualitatively similar results (*Figure 6G*), slightly underpredicting the effect of 4 ss intervention and predicting 5 ss and 5 ss +7 ss interventions would lead to a higher defect rate than observed. This model assigned almost all the contribution to the 5 ss stage (*Figure 6I*). Omitting 3 ss - 5 ss contributions led to clearly incorrect predictions of the effect of 6 ss intervention (*Figure 6J*) and omitting 3 ss - 6 ss contributions led to a model that was not able to make any match to observations (*Figure 6M*). Overall, these results emphasised the crucial contribution of the 4 ss-6 ss stages, and possible contribution also of the 3 ss stage.

## Earlier anterior detection of *dand5* asymmetric expression supports time-window and anterior flow contribution to symmetry breaking

We have experimentally established the most sensitive period for symmetry breaking at the KV from 4 to 6 ss, which is backed up by our theoretical model, however, left-right asymmetry in *dand5* expression was only detected at 8 ss using traditional in situ hybridization methods (*Lopes et al., 2010*), which have limited resolution. The number of *dand5* mRNA transcripts on each side of the KV over time has not been quantified in previous studies due to the lack of spatial resolution in quantitative methods such as RT-qPCR and RNA-seq (*Tavares et al., 2017*; *White et al., 2017*). Therefore, it is unclear when and which cells initiate *dand5* mRNA decay. Furthermore, it has not been explored whether these cells are exposed to different flow velocities. Moreover, there have been no 3D evaluations of the dynamics of *dand5* transcripts over time at the KV.

To improve the spatial resolution and specificity of *dand5* mRNA transcript detection, we established a protocol using single-molecule fluorescence in situ hybridization (smFISH) using Tg(sox-17:GFP) embryos to delimit the KV. Unlike traditional in situ hybridization methods, smFISH allows the detection of individual mRNA transcripts with high sensitivity and spatial resolution.

We performed a time course from 3 to 8 ss and detected the total number of spots on the left and right sides of the KV at each developmental stage (*Figure 7A and C*). We also performed the same protocol on 8 ss *dand5* homozygous embryos, that have no transcripts, to validate the spot-detection algorithm (*Figure 7B*). We found symmetric expression of *dand5* in both 3 and 5 ss (*Figure 7C*). At 6 ss, we detected a right-sided asymmetry (p-value = 0.0012) that persisted throughout 7 ss (p-value = 0.0015) and 8 ss (p-value = 0.0347), making 6 ss the earliest *dand5* L<R asymmetric expression reported (*Figure 7C*). We also compared the number of detected transcripts on the left and right anterior sides of the KV from 3 to 8 ss and found that *dand5* expression was symmetrical at 3 and 5 ss, but a right-sided anterior asymmetry was established at 6 ss (p-value = 0.0378) and sustained at 7 ss (p-value = 0.0467) and 8 ss (p-value = 0.0197) (*Figure 7D*). Together, these results indicated that *dand5* asymmetry is established at 6 ss, which is 1 hr earlier than the previously reported timing

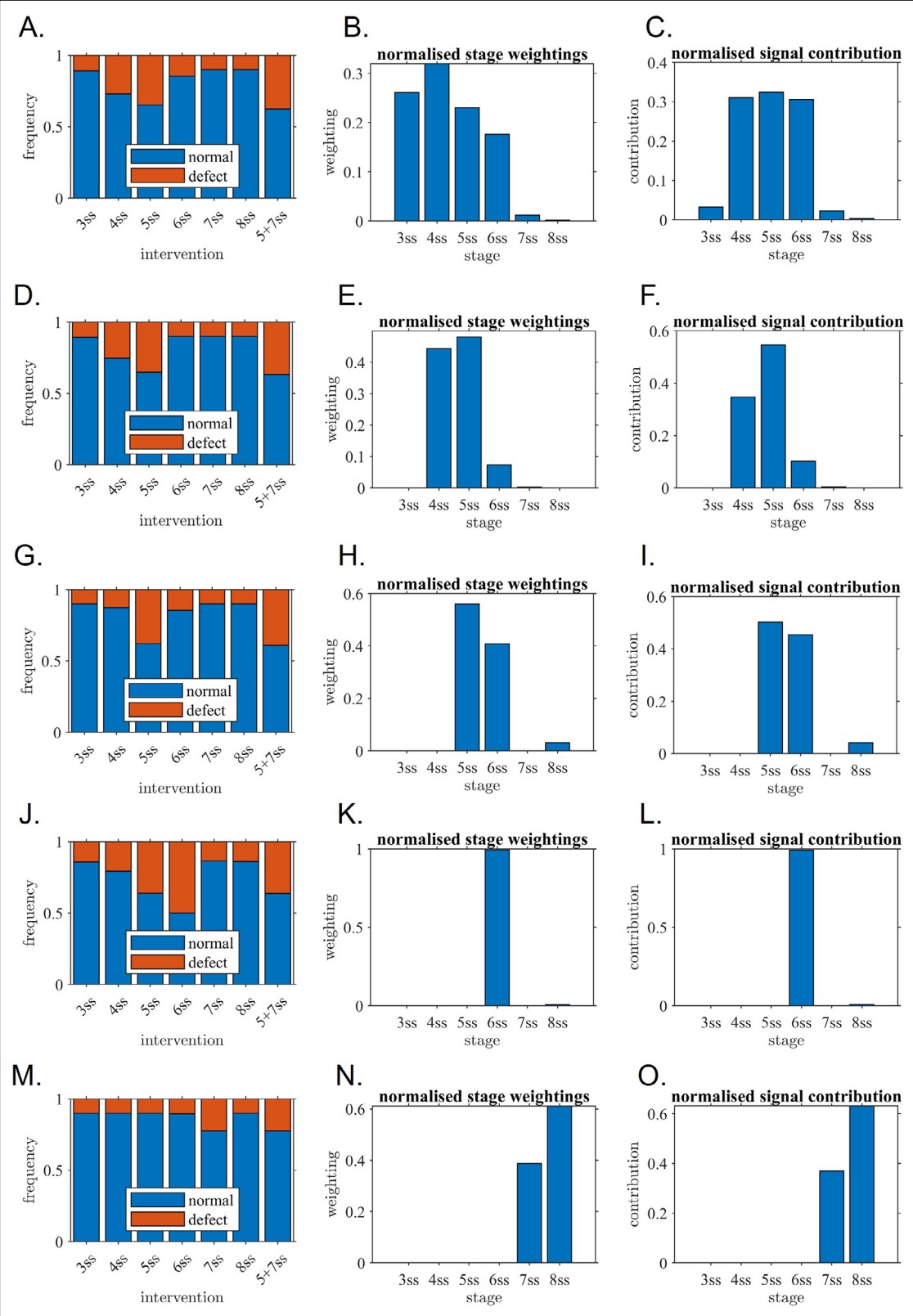

**Figure 6.** Mathematical modelling confirms the critical role of one-hour interval confirming the experimental observations. Outcome of model fitting to experiment, associated stage weightings (sensitivity of KV to its own flow) and signal contribution (sensitivity multiplied by flow strength) for (**A–C**) Unrestricted model, (**D–F**) model with 3 ss contribution omitted, (**G–I**) model with 3–4 ss contributions omitted, (**J–L**) model with 3–5 ss contributions omitted, (**M–O**) model with 3–6 ss contributions omitted. The restricted models are included to assess alternative explanations for the data in which

*Figure 6 continued on next page*

*Figure 6 continued*

later stages still make a significant contribution. (**A, D, G, J, M**) fitted model outputs of rates of normal and abnormal situs for direct comparison with experiment; (**B, E, H, K, N**) fitted stage weightings (sensitivity of KV to its own flow) where tall bars mean 'relatively more sensitive' in arbitrary units, (**C, F, I, L, O**) signal contribution (sensitivity multiplied by flow strength) where tall bars mean 'relatively more contribution'. The fitted restricted models for (**G–I, J–L and M–O**) do not fit the experiment well which confirms that they are unlikely to represent what is occurring in the KV. The fitted models for (**A–C, D–F and G–I**) have a much better fit and confirm that 5ss makes a major contribution, as do 4ss and/or 6ss. ss: somite stage.

The online version of this article includes the following figure supplement(s) for figure 6:

**Figure supplement 1.** Mixed effects model fits for the distributions of anterior angular velocity to (**A**) data for sham intervention, (**B**) data for suction intervention at 5 ss.

**Figure supplement 2.** Anterior angular velocity distributions for all modelled experimental interventions.

(*Lopes et al., 2010*). Interestingly, we found that *dand5* mRNA reduced expression on the left-side was mainly restricted to the anterior region of the KV, where flow from right-to-left has stronger angular velocity, suggesting that the left-anterior cells are the first to perceive the flow and show a difference in *dand5* expression. Supporting this idea, Yuan et al. reported significant higher anterior left intraciliary calcium oscillations as an initial response to fluid flow sensing dependent on Pkd2 (*Yuan et al., 2015*). The dependency of Pkd2 was recently confirmed by both *Djenoune et al., 2023* and *Katoh et al., 2023*, in zebrafish and mouse, respectively.

## Discussion

By challenging the zebrafish left-right organizer (LRO) with a new form of manipulation we tested the physiological time limits for symmetry breaking. With this information we highlighted the most sensitive developmental stages. We narrowed the LR time-window for symmetry breaking to 1 hr, peaking at 5 ss. Next, we focused on 5 ss and repeated fluid extractions to study flow recovery dynamics from 6 to 8 ss, carefully analysing progression of midplane area, fluid angular velocity components and flow directionality. Most embryos succeeded to display a correct LR organ *situs* and did so by quickly restoring fluid and flow anterior angular velocity. In embryos with LR defects we found a general decreased anterior angular velocity, a smaller difference between anterior-posterior angular velocity and an abnormal flow directionality in anterior sections of the LRO.

(*Yuan et al., 2015*) showed that in unperturbed embryos the intraciliary calcium signal intensity peaked on the left side of the LRO around 3 ss. This period coincides with the first defects we observed upon depletion of the LRO fluid content. Nevertheless, results and modelling showed that these younger embryos may have greater tolerance to challenges and higher chances of recovering the normal LRO fluid and flow parameters and subsequent correct LR pattern than those extracted at 5 ss. Therefore, physiologically, our results highlighted the most susceptible timing to the fluid manipulation challenge was at 5 ss, which does not mean that intraciliary calcium transients do not occur before or cannot be triggered by using optical tweezers, as recently shown by *Djenoune et al., 2023* in zebrafish and by *Katoh et al., 2023* in mice. Other studies showed that a process of LRO remodelling occurs between 4 and 6 ss resulting in a biased ciliary distribution to the anterior region (*Kreiling et al., 2007*, *Okabe et al., 2008*; *Wang et al., 2011*). It is also at this period that we previously observed an increased number of motile cilia at the expense of immotile cilia, in robustness of the fluid flow and in establishment of characteristic patterns of fluid dynamics, such as the higher flow speed in the anterior dorsal pole (*Smith et al., 2014*). We showed that if the fluid extraction is made until 6 ss the system senses it and there is a response that may compensate the insult but if the extraction is made at 7 ss the LRO does not sense it anymore. This heightened the need of identifying the mechanism that senses the challenge in the relevant short time-interval.

We therefore proceeded to assess the effect of replacing extracted fluid with either buffer, which simply dilutes the KV contents, or buffer with added methylcellulose, which also increases viscosity and serves here as a positive control. Simple dilution did not significantly affect situs whereas increasing viscosity dramatically increased rates of right situs. Statistical analysis of flow data showed that increasing the viscosity via injection is associated with very strong reduction in angular velocity, a strong reduction in anterior-dominance of the flow, and modestly faster flow on the right compared with the left. By contrast, dilution which does not increase viscosity is associated with a modest reduction in anterior-dominance and a modest reduction in flow acceleration over time. The strong effects

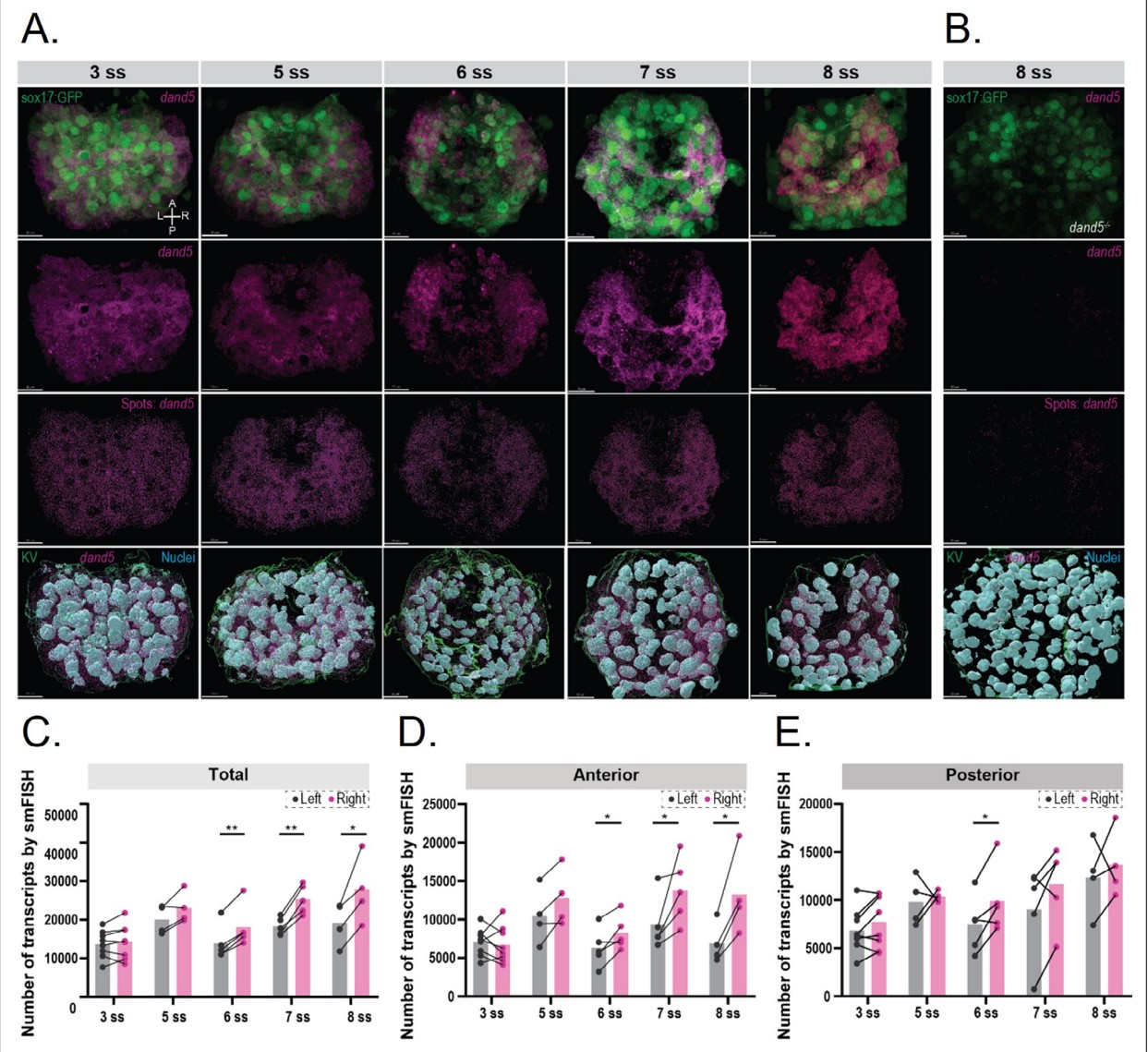

**Figure 7.** 3D smFISH and spatial analysis of *dand5* mRNA transcripts in the KV. (**A**) Time-course 3D analysis of *dand5* mRNA transcripts by smFISH in Tg(sox17:GFP) embryos at 3, 5, 6, 7, and 8 somite stage (ss). 3D representations of *dand5* mRNA (magenta), nuclei (cyan), and sox17:GFP (green) are shown. The KV surface (green) was used to mask both the *dand5*-Atto 565 and DAPI channels to remove nuclear staining. Spot detection was used to detect *the dand5* cytoplasmic transcripts. Axes are indicated. Scale bar = 15 μm (**B**) The spot detection algorithm was validated in 8 ss *dand5⁻/⁻* embryos raised on the *Tg(sox17:GFP)* background. (**C–E**) Quantification of differences in the number of *dand5* mRNA transcripts detected on the left and right sides of the KV; total number (**C**), at the anterior (**D**), and posterior (**E**) sides of the KV. Gray and magenta represent the left and right sides, respectively. Paired t-tests were used. The bars represent the mean values, and the dots represent individual embryos. Asterisks denote statistical significance: * corresponds to a p-value <0.05, and ** indicates a p-value <0.005.

The online version of this article includes the following source data for figure 7:

**Source data 1.** Quantification of cytoplasmatic *dand5* expression by smFISH in the LRO left, right, anterior and posterior halves.

appear to explain the effect of viscosity on situs. We therefore concluded that the sensory mechanism detects flow properties, as angular velocity and viscosity but seems to be insensitive if we quickly replace the fluid and keep flow dynamics fairly constant.

In the presence of a low Reynolds number, when viscosity dominates and inertia is negligible, a sensory system to be efficient must be very sensitive, as suggested in *Shinohara et al., 2012*. The most plausible mechanosensory system in the zebrafish LRO involves Pkd1l1 and Pkd2, both present in the LRO cilia as shown by immunofluorescence (*Roxo-Rosa and Santos Lopes, 2020*; *Jacinto et al., 2021*) and by recent functional studies for Pkd2 by *Djenoune et al., 2023*. Although it has not

been possible to demonstrate that these two Polycystin proteins function together as a mechanosensory complex in zebrafish, there is evidence to be the case in the mouse node (*Field et al., 2011*; *Mizuno et al., 2020*; *Katoh et al., 2023*). Moreover, patients with mutations in PKD1L1 or in PKD2 present laterality phenotypes such as *situs inversus*, heterotaxia and congenital heart disease (*Bataille et al., 2011*; *Le Fevre et al., 2020*).

Live quantifications of cilia motility along time, using two-photon microscopy (*Tavares et al., 2017*; *Ferreira et al., 2017*) attested that in the first stages after the formation of the LRO, there are many more immotile cilia (around 94%) than motile, but then, a gradual loss of cilia immotility occurs during development. The present study showed that in all embryos that developed correct LR 'Sham' and 'No defects' groups; scored at 6 ss, very few immotile cilia were present (9 on average) in a total average of 43 cilia (n=13 embryos), which is only 20,9% of the total LRO cilia. So, according to *Ferreira et al., 2017*, this fact brings to light that perhaps immotile cilia number is not a crucial factor, at least in zebrafish. A more recent hypothesis for the sensory system in the zebrafish embryo postulates that cilia themselves may sense their own movement (*Ferreira et al., 2019*). In fact, *Yuan et al., 2015* filmed intraciliary calcium oscillations (ICOs) in the zebrafish LRO motile cilia in action and showed that ICOs were abrogated in knockdowns for Pkd2 or in cilia motility mutants. The motile cilia self-sensing model becomes appealing in view of the flow response to increased viscosity. Motile cilia pattern and power are known to differ at increasing viscosities (*Kikuchi et al., 2004*; *Wilson et al., 2015*; *Gallagher et al., 2019*), which could potentially be detected by a ciliary mechanosensitive transduction channel, like in the fish lateral line where stereocilia bending allows sensing the velocity and direction of water flow (*Asadnia et al., 2016*). In summary, we suggest that a ciliary mechanosensory system, such as Pkd1l1/Pkd2 could be sensitive to a sustained flow angular velocity.

The role of Pkd2 and calcium in LR symmetry breaking is now unquestionable (*Katoh et al., 2023*; *Djenoune et al., 2023*). Both these recent studies elegantly suggest a mechanosensing component in LR symmetry breaking, totally independently from our work. A previous study by *Delling et al., 2016* contradicted the idea that crown cells LRO cilia were good calcium-responsive mechanosensors. Conversely, work from Hamada's lab *Minegishi et al., 2021* demonstrated that LR asymmetric intraciliary calcium transients can be in fact detected at the node. Mizuno and colleagues further revealed that discrepancies between the two studies relate to the culture conditions used, which led calcium signals to be missed in the work by *Delling et al., 2016*. Moreover, Minegishi et al. unveiled that in the mouse model, the 3'-UTR of *dand5* mRNA responds to the direction of fluid flow in a Pkd2-dependent manner via stimulating Bicc1 and Ccr4-Not. Together, these molecular players seem to mediate *dand5* mRNA degradation specifically on the left side of the mouse node and *Xenopus* gastrocel roof plate (*Maerker et al., 2021*). Recent work by *Katoh et al., 2023* provides the first evidence-based mechanism for how Pkd2 channels may be preferentially activated by the flow on the left-side of the mouse node. Katoh et al. show asymmetric enrichment of Pkd2 protein in the ciliary membrane of each crown cell cilium with reference to the midline.

We and others showed that in zebrafish there is a counterclockwise flow direction and an anterior-posterior positive ratio in angular velocity, which causes a stronger right-to-left flow anteriorly compared to a slower flow from left-to-right posteriorly. Our fluid extraction experiments highlighted these two properties must be met for zebrafish to recover and succeed in symmetry breaking in a defined time-window. So, cells at the anterior-left pole are likely to experience stronger flow as showed by radial velocity denoting flow directionality towards the membrane was enhanced in this region.

Concomitantly, performing cellular resolution smFISH showed anterior left sided cells are the first to show a reduction in *dand5* expression as early as 6 ss. How the anterior right to left stronger flow is mechanistically and molecularly translated into anterior-left reduction of *dand5* mRNA remains unclear in zebrafish. If *dand5* mRNA decay is the process involved, then this is a relatively fast process (measured by *Katoh et al., 2023*), which physiologically should happen straight after the sensitive time-window we defined here. So, the observed left-anterior reduction of *dand5* expression at 6 ss matches the correct spot and timing. Therefore, in a consistent timeline, most zebrafish embryos start to display a LRO lumen by 2 ss (*Oteíza et al., 2008* and our own lab), then according to our data, reach a physiological window for breaking symmetry at 4 ss that persists until 6 ss, culminating with the first left-sided *dand5* mRNA reduction at 6 ss.

According to a recent evolutionary developmental biology publication, there are five conserved genes that are predicted to be involved in symmetry breaking in vertebrates with ciliated LROs, we can only speculate that perhaps CIROP and/or Metalloproteinase21 function, predicted to be downstream of flow, may further explain the left-sided *dand5* mRNA decrease (*Szenker-Ravi et al., 2022*).

After fluid extraction, the time lag and complexity of the fluid flow recovery process motivated the development of a mathematical model of flow, its variability, and the effect of experimental interventions, with the aim of determining the sensitivity of KV to its own flow field through fitting experimental data. The core idea of the model is to represent symmetry breaking as a random process which is biased by flow. The analysis took account of natural variability in the flow field and its modification by intervention, with the biasing effects of each stage to be determined through fitting, and for a comparison to be made against alternative models in which only later developmental stages are important. This analysis confirmed the critical importance of 4 ss to 6 ss in symmetry breaking, and the limited role for flow during 7 ss and 8 ss. The process of constructing the model also emphasised that intervention has two effects: the transient gross reduction in flow associated with removal of the entire luminal fluid, but also a greater variance of flow following recovery – hence observed defects are as much a consequence of a proportion of embryos making a below average recovery, as the interruption itself.

Finally, we need to discuss whether our experiments exclude the potential contribution of signalling morphogens or EV secretion to LR establishment. Literature on EV secretion rates is scarce but we found an estimate of 60–80 EVs per cell per hour is expected, depending on the cell type (*Agarwal et al., 2015*; *Chiu et al., 2016*). According to *Ferreira et al., 2017* in a simulated transport of signaling molecules in the Kupffer's vesicle between 9–14 ss, mixing would take only 8 s to occur. At 5 ss, at the peak of physiological symmetry breaking, we estimate that mixing would take longer to occur because flow is weaker than at 9–14 ss, but nevertheless it would be fast, perhaps taking a few minutes. Thus, according to this line of thought, it is still possible that secretion of morphogens or EVs can occur between fluid extraction at 5 ss and the limit of the responsive time-window by 7 ss (1 hr interval). The only problem with this hypothesis is that all fluid extraction experiments (n=33) resulted in *dand5* defects at 8 ss, except when we replaced the fluid with physiological buffer. If a secreted molecule was being continuously replenished for 1 hr, we would expect absence of *dand5* defective phenotypes in all extraction experiments since the extractions were performed at 5 ss and the fixation for *dand5* conventional in situ hybridization was done at 8 ss, meaning 2 hr later, allowing enough time for recovery of any secreted factor within the competent time-window. Therefore, our results suggest the zebrafish LRO is more sensitive to mechanical properties for breaking symmetry.

## Methods

### Fish husbandry and strains

Zebrafish, *Danio rerio*, were maintained at 25 °C or 30 °C and staged as described elsewhere (*Kimmel et al., 1995*) according to the number of somites. The following zebrafish lines were used for this work: AB and Tg(sox17:GFP)s870 (*Chung and Stainier, 2008*). Procedures with zebrafish were approved by the Portuguese authority DGAV (Direção Geral de Alimentação e Veterinária) *title: "Estabelecimento do eixo direita-esquerda no organismo modelo peixe-zebra".*

### LRO manipulation setup for liquid extraction

LRO manipulations of fluid extraction were performed from 3 to 12 somite stages (ss). Zebrafish embryos were individually chosen at the developmental time correspondent to each somite-stage. Furthermore, only embryos that recovered LRO fluid were used throughout the study. Individually embryos were hold from the anterior dorsal side of the body using a holding pipette while a second pipette was used to penetrate the LRO from the posterior dorsal side. After extracting the LRO liquid the microinjection pipette was withdrawn, and embryos were incubated and let to develop. Collectively 173 embryos were manipulated along the different stages. Embryos were let to develop at 28 °C and were later characterized according to *dand5* expression pattern or to organ laterality. The micromanipulation system, as illustrated in *Figure 1*, was composed of an inverted microscope (Andor Revolution WD, Oxford Instruments; or Nikon Eclipse Ti-U) with a 10 x Plan DL air objective, a 60 x Plan Apo VC water immersion and a 100 x Plan Fluor oil immersion objective (Nikon), a set of

two motorized axis manipulators (MPC-385–2, Sutter Instruments) placed on both sides of a culture dish for moving pipettes to the desired positions, a CMOS camera (Monochrome UI3370CP-M-GL, Imaging Development Systems GmbH; or high speed FASTCAM MC2 camera, Photron Europe, Limited) to capture images in the microscope field of view, and a set of two CellTram's (CellTram Vario, FemtoJet, Eppendorf). Each respective microinjector are connected to a holding micropipette for immobilizing the zebrafish embryo, and injection micropipette that allows fluid manipulation from or to the Kupffer's vesicle (KV). All the setup components were constructed over an optical table with vibration isolation (ThorLabs) to minimize the effects of any residual vibrations or disturbances introduced during the embryo manipulation procedure.

## Immobilizing embryos

The embryo holding step was based on immobilization of the embryo by a pipette positioned on its left side in order to allow the injection micropipette to be introduced from the right side through the dorsal tissue into the KV. Of the several positions tested immobilization by the zone dorsal to the developing head of the embryo was the most effective. Unlike the yolk, the back of the head offered a resistant tissue with a shape that easily adapted to the diameter of the holding pipette. At the same time this setup allowed the KV to be aligned with the microinjection needle field of action. The creation of a chamber for easy handling of the injection micropipettes while keeping the specimen in an aqueous solution was crucial for the success of this procedure. Using silicone grease (Dow Corning) a rectangular coating was outlined on a coverslip. Next, a sufficient amount of embryo medium E3 was carefully pipetted on the coverslip forming a liquid pond. This setup contained zebrafish embryos in an aqueous environment with very low physical impediment, allowing a high range of contact angle for micropipette manipulation to happen.

## Pipette forging for micropipette extractions

During its developmental time period the LRO or KV will migrate from 70 to 120 μm below the tail mesodermal cell layer (*Kimmel et al., 1995*). Consequently, vesicular fluid extraction and compound microinjection procedures require resistant sharp capillaries capable of piercing through the tail tissue dorsal to the KV. Micropipette tip size and shape of the taper are crucial factors determining if a micropipette will effectively impale the embryo tissue. Small tips and uniform tapers have the great advantage of causing less cell damage (Sutter Instrument, pipette cookbook, 2008) http://www.sutter.com/contact/faqs/pipette_cookbook.pdf. Likewise, a shorter tapered and very stiff micropipette is needed to penetrate very tough and rigid membranes.

Aluminosilicate glass capillaries are known to have an increased hardness and ability to form small tips compared with its borosilicate counterpart, allowing to form stiff longer tapers important for reaching the KV and avoiding unwanted tissue damage. Taking this in consideration, microinjection pipettes (Aluminosilicate, inner diameter 0.68 mm, outer diameter 1.00, length 10 mm, without inner filament) were designed using a custom-made program in a horizontal filament puller (P-97 Micropipette Puller, Sutter Instrument). Reduced cone angle and slender taper can be achieved using a 2-loop puller program.

The resulting pulled pipettes were often sealed or have a very narrow tip (<1 μm). Shaping the tip to the desirable size was important to avoid clogging of the pipette and deliver a sharp cut when impaling the embryo tissue. Therefore, the tips of micropipettes were shaped to small apertures (<5 μm) under the microscope. Using the motorized manipulators, the tips were carefully cut-opened against a polished borosilicate capillary. This allows to have a balance between micropipette longevity and minimal tissue damage during embryo manipulation. For better results reducing the amount of mechanical stress caused by the needle injection, a micropipette beveller (BV-10, Sutter Instrument) was used to increase sharpness of the pipette tips. This additional step was not always carried out due to the significant time increase in preparation of each micropipette.

## Holding pipette

Since our organism of interest is immersed in an aqueous non-viscous solution, oscillations of the zebrafish embryo are a major concern during micromanipulation. Thus, special care should be taken to assure a rigid stable connection between the embryo tissue and the tip of the micropipette. To achieve this a holding pipette (typically positioned on the left side of the preparation), immobilized

the specimen while an opposing injection micropipette (on the right side) was introduced into the embryo tail.

For forging holding pipettes, thin wall borosilicate capillaries (inner diameter 0.75 mm, outer diameter 1.00 mm, length 10 mm, without inner filament) were used. Ideal holding pipettes have a flat straight break at the tip with large inner and outer diameter to provide better attachment and support to the zebrafish embryo. Fire-polishing was then performed to create a smooth surface to interface with the embryo tissue without damaging it, and produce an inner diameter (around 0.50 mm) best suited to hold the embryo as described in *Brown and Flaming, 1974*, where a common Bunsen burner or a micro forge (MF-900, Narishige) were used to fire-polish the holding pipette.

## Cilia number, length and anterior-posterior distribution in fixed samples

To ensure that low damage is caused during embryo manipulation we further quantified KV cilia number and length upon fluid extraction. After manipulations at different days a subsample of embryos were immediately fixed in PFA 4% and whole-mount immunostaining for acetylated α-tubulin was performed to label KV cilia as previously described (*Lopes et al., 2010*). Antibodies used for immunostaining were mouse anti-acetylated alpha-tubulin (1:400; Sigma) and Alexa Fluor 488 (Invitrogen; 1:500). Cilia count and length measurements were performed either manually or through semiautomated detection in IMARIS software program.

## Evaluating motile / immotile cilia distribution by live imaging

Subsamples of embryos collected at different days were imaged. Procedure followed the method described in *Tavares et al., 2017*. Live imaging was performed in a Zeiss LSM 710 confocal microscope with an Olympus 40 x water immersion lens (NA 0.8) at room temperature. Cilia identification was performed either manually or through semi-automated detection in IMARIS software program (Bitplane, UK).

## Live imaging of fluid flow in the zebrafish left-right organizer

For the LRO fluid extraction experiments, embryos collected at different days were maintained between 28°C–30°C until 5 ss, then mounted in a 2% (w/v) agarose mold and covered with E3 medium, and filmed at 25 °C. To evaluate the fluid flow and LRO inflation recovery dynamics, the LRO was followed along development, from 6 to 8 ss in a sample of embryos manipulated at 5 ss (n=16). In parallel, embryos (n=6) in which the LRO was punctured by a micropipette, without further liquid extraction, were used as controls ('Sham' group). Imaging was performed on a light microscope (Nikon Eclipse Ti-U inverted microscope), under a 60 x/1.2 NA water immersion or 100×/1.30 NA oil immersion objective lens, by a FASTCAM-MC2 camera (Photron Europe Limited, UK) controlled with Photron FASTCAM Viewer software. Using the ImageJ plugin Measure Stack, the LRO was delineated, and its luminal area was measured in all focal planes. LRO particles were tracked using the ImageJ plugin MTrackJ as previously reported (*Sampaio et al., 2014*).

## Evaluation of heart and gut laterality

The same embryos used above were used for evaluation of heart jogging at 30 hr post fertilization (hpf) using a stereoscopic zoom microscope (SMZ745, Nikon Corporation). These embryos were then allowed to develop at 28 °C and at 53 hpf embryos were fixed and processed for *foxa3* in situ hybridizations to assess gut laterality (*Sampaio et al., 2014*), or readily scored if a sox17:GFP transgenic line was being used for the respective assay. We could then pair the heart *situs* with gut *situs* for each treatment and attribute an embryo *situs*. Final organ *situs* score was attributed regarding the combined heart and liver lateralization status: *situs solitus* – normal organ patterning (left heart and liver), *situs inversus* - organs positioned opposite to normal positions (right heart and liver), and heterotaxia – comprising all the other laterality defects variants. For the LRO fluid extraction assay a total of 1223 WT embryos and 173 manipulated embryos were characterized according to organ *situs*.

## RNA in situ hybridization and immunofluorescence

Whole-mount in situ hybridization and immunostaining was carried out as before (*Lopes et al., 2010*) Digoxigenin RNA probes were synthesized from DNA templates of *dand5* (*Hashimoto et al., 2004*).

## SmFISH probe synthesis and protocol

Probes for single-molecule fluorescent in situ hybridization (smFISH) targeting the coding, 5'-UTR, and 3'-UTR regions of the zebrafish *dand5* gene were designed using the Stellaris Probe Designer and ordered from IDT. The smFISH probes were labeled with amino-11-ddUTP and Atto565 NHS ester dye using a protocol adapted from that described by *Gaspar et al., 2017*. The labeled oligos were purified using a Zymo Oligo Clean & Concentrator kit and the concentration was measured using a NanoDrop 2000 spectrophotometer.

The smFISH staining protocol was adapted from *Oka and Sato, 2015*, and *Lord et al., 2021*. First, the embryos were fixed in 4% paraformaldehyde (PFA) at 4 °C ON, with ten embryos per Eppendorf tube. Next, the embryos were dehydrated after a quick wash in PBSTw (1 x PBS +0.1% Tween 20). The first step involved a 5 min wash in 50% MeOH and 50% PBSTW RT, followed by a 5 min wash in 100% MeOH. The embryos were then stored at –20 °C until staining. On the first day of the protocol, the embryos were rehydrated in a series of MeOH and PBSTw solutions (50% and 100% PBSTw) before the manual removal of the chorion. The embryos were then incubated for 30 min at 30 °C in 300 µL of pre-hybridization buffer (10% formamide, 2 x SSC, 0.1% TritonX-100, 0.02% BSA, and 2 mM ribonucleoside-vanadyl complex [NEB]). This was followed by ON incubation in the dark at 30 °C with probes diluted 1:100 in 100 µL of hybridization buffer (10% dextran sulfate dissolved in pre-hybridization buffer) and anti-GFP (1:500; Invitrogen) diluted in hybridization buffer with the probes. After staining, the embryos were washed twice for 30 min each at 30 °C in a hybridization washing solution (10% formamide, 2 x SSC, and 0.1% Triton X-100), followed by a brief rinse in a solution of 2 x SSC and 0.1% Tween-20, and incubated for 20 min at 30 °C in 0.2 X SSC. The embryos were then incubated for 2 h at room temperature in the dark with anti-rabbit Alexa Fluor 488 secondary antibody (Invitrogen, 1:500), followed by six washes with PBSTw for 5 min each. Embryos were then incubated with DAPI for 15 min at RT in the dark, followed by two washes in 2 x SSC and 0.1% Tween-20 for 5 min each. If not mounted immediately, the embryos were stored in 2 x SSC at 4 °C in the dark.

## SmFISH imaging and analysis

Imaging of the embryos was performed using a Zeiss LSM-980 Airyscan Confocal Microscope equipped with Airyscan SR detectors and controlled by Zen software version 3.3. The z-stack acquisition was achieved using a 40 X water immersion objective and a 1.8×zoom factor, with images obtained at 0.18 µm intervals. 3D datasets were analyzed using the software IMARIS v. 9.8.2. The first step was to align the axes with the notochord, visible in the DAPI channel, by rotating the image and applying the 'Free Rotate' image processing setting. A surface representation of the KV was then generated by manually tracing it in the AF 488 channel. The surface was then used to mask the Atto565 and DAPI channels to eliminate voxels outside the surface. Next, an automatic detection setting was applied to the masked DAPI channel to create a surface representation of the nuclei, which was then used to mask the Atto565 channel and to remove the fluorescence signal from the nuclei. Finally, the KV was divided into two regions of interest, and automatic spot detection was applied to these ROIs to quantify the number of cytoplasmic *dand5* mRNA transcripts. An XZ clipping plane was used to select only the spots on either the anterior or posterior side, allowing the extraction of the anterior and posterior number of spots on both sides of the KV.

## Flow speed, angular velocity, and particle directionality

The flow speed, angular velocities and velocity components were calculated by customized scripts in the program environment R (Supplementary data). The centre of the LRO was used as reference point of a polar coordinate system in which angles are expressed in radians. To study the flow dynamics near the LRO apical membrane, tracked points near the centre (corresponding to a circle with half of the LRO radius) were removed, as angular quantities are poorly defined along the axis of rotation, close to the centre of the LRO. Angular velocity represents the effective circular flow magnitude of a particle moving between two consecutive time frames. The centre of the KV was used as reference point of a polar coordinate system in which angles are expressed in radians. Instantaneous angular velocity (rad/sec) was calculated by dividing particle angle change by the time between consecutive image frames (0.2 s). Particle directionality quantification at any given point was obtained by dividing the LRO in 30 degree sections and calculating the change of particle angle between two consecutive

time frames, where the axis of the reference is the position of the particle in the first time point. The particle directionality values range from 0 to 2π.

## Fluid dilution and viscosity manipulation in the zebrafish LRO

The KV fluid dilution assay was performed by aspirating the vesicular fluid into a micropipette previously filled with Danieau's solution 1 x (58 mM NaCl, 0.7 mM KCl, 0.4 mM MgSO4, 0.6 mM Ca(NO3)2, 5.0 mM HEPES pH 7.6) diluted in 10,000 MW rhodamine-dextran solution (1:4; Sigma-Aldrich). The mixed solution was injected again into the KV and only embryos with rhodamine-positive KV's were selected for the experiment. For viscosity manipulation, the micropipette was filled with Danieau's solution 1 x containing 1.5% (w/v) of methylcellulose (M0555, Sigma). Aspirated KV fluid and methylcellulose were then injected again into the KV lumen until reaching the KV volume observed before the manipulation.

## Statistical analysis

A Fisher's exact test performed with RStudio was used to compare frequency of left-right defects and *dand5* expression patterns between WT and manipulated embryos. To assess potential differences in KV area, cilia number, cilia length between WT and manipulated embryos Student's t-test was used. To assess area changes per somite Mann-Whitney U test was used. To assess cilia motility differences between anterior and posterior regions Fisher's exact test was used. Kolmogrov-Smirnov test was used from comparing particle trajectory distribution between two groups. Angular velocity data for tracks imaged in each of the suction and injection experiments was analysed via linear mixed effects regression to characterise how angular velocity varies across LRO and over time, how these velocity field patterns are altered due to intervention, and how they are reflected by different embryo fates. Fixed effects were group (injection experiment: sham, buffer injection, methylcellulose; suction experiment: sham, no-defects, defects), somite stage, normalised posterior-anterior axis coordinate, normalised left-right axis coordinate, and interactions of group with somite stage, posterior-anterior and left-right coordinates. Random effects were included to account for heterogeneity between embryos in their response to intervention. The regression was performed specified in Matlab (Mathworks) notation as,

Suction experiment:

AAV ~1 + Group*LR +Group*PA +Group*Time + (1+Group | EmbryoID)

Injection experiment:

AAV ~1 + Intervention*LR +Intervention*PA +Intervention*Time + (1+Intervention | Embryo ID)

Fitting was carried out with the command fitlme, with effects reported in the main text where p<0.05 and interpreted based on the sign and magnitude of the fitted coefficient.

## Mathematical modelling

Mathematical models of anterior angular velocity, its disruption by suction intervention, and the signal integration process, were implemented as custom scripts (*Smith, 2023*) in Matlab (R2021a, Mathworks, Inc, Natick MA).

## Statistical models for anterior angular velocity

The key hypothesis of the model is that symmetry breaking is produced by anterior angular velocity AAV(t) at somite stage $t$ integrated from 3 ss to 8 ss, weighted by the a priori unknown sensitivities W(t) at each stage. As the aim here is only to provide a good fit to the flow velocity data as input to the signal transduction model, the precise mathematical form of the model is not relevant. Mixed effects regression was used to approximate anterior angular velocity data and its variability for 5 ss-intervention and sham intervention using the Matlab function nlmefit (nonlinear mixed-effects estimation).

AAV following intervention at 5 ss was modelled using the nonlinear function

$$AAV_{interv_{5ss}}\left(t\right) = \varphi_1 \left(\frac{1}{1+e^{-0.9(t-5-\varphi_2)}} - \frac{1}{1+e^{0.9\varphi_2}}\right) + \frac{\varphi_3}{1+e^{-8(t-5.5)}}$$

The coefficients 0.9 and 8 were chosen to provide a smooth increase from zero, and the parameters $\varphi_j$ were determined by fitting. Mixed effects regression produces estimates for the fixed effect (central

tendency in the population) and the standard deviation of the random effect quantifying variability in the population; to maintain parsimony only the final parameter $\varphi_3$ was modelled with a random effect. Estimates for the fixed effects were $\varphi_1 = 0.778 \pm 0.679$, $\varphi_2 = 0.969 \pm 2.562$ and $\varphi_3 = 0.126 \pm 0.249$. The random effect for $\varphi_3$ was estimated to be 0.169. While standard errors are relatively large, the purpose of the fitting process was not the parameter values per se, rather to obtain a representative model of the recovery process. The resulting model is shown in Figure S11A.

AAV following sham intervention at 5ss was modelled using the linear function

$$AAV(t) = \varphi_1 + \varphi_2(t-5)$$

The intercept parameter $\varphi_1$ was modelled with a normally-distributed random effect. Estimates for the fixed effects were $\varphi_1 = 0.356 \pm 0.063$ and $\varphi_2 = 0.0362 \pm 0.0283$; the random effect for $\varphi_1$ was estimated to be 0.0378. The model is shown in Figure S11B.

In the case of sham intervention, based on linearity of Stokes flow, AAV at $3ss - 5ss$ was rescaled based on observed AAV at 6 ss grounded on previously-observed data on motile cilia percentage, specifically 5.57% at 3 ss, 42.9% at 4 ss, 62.1% at 5 ss and 76.3% at 6 ss (so that $AAV(3) = (5.57/76.3)\,AAV(6)$ for example). In the cases of intervention at 3 ss or 4 ss, the function $AAV_{interv_{5ss}}(t)$ was shifted in time so that $AAV_{interv_{3ss}}(t) = AAV_{interv_{5ss}}(t+2)$, then for $3 \le t \le 5$ rescaled based on the motile cilia percentages given above.

The modelled AAV distributions for every case are given in Figure S12.

## Signal integration and symmetry breaking

Symmetry breaking signal $S$ is modelled as a weighted sum at each somite stage,

$$S = \sum_{t=3}^{8} AAV(t)\,W(t)$$

where the parameters $W(t)$ are the sensitivity weighting of each stage $t = 3, \dots, 8$. The weightings are independent of the experimental situation and so are determined simultaneously across all experiment series.

A given embryo is assumed to break symmetry normally at a baseline rate of 50% in the absence of any signal at all, and at a rate of 90% if the signal exceeds a threshold (taken to be 1 in arbitrary units).

When sampling over 1000 embryos, the abnormality rate therefore lies between 50% and 90% depending on the distribution of signal $S$. This abnormality rate is used as the success probability for the binomial model used for model fitting.

## Signal integration model fitting

There parameters W(t) are determined through maximum likelihood estimation combined with global search, treating each experiment series with a binomial model, with success probability determined from S as described above. The log-likelihood is then maximized through global optimization, by applying the MultiStart and GlobalSearch functions (Matlab Global Optimization Toolbox). A uniform initial guess $W(3) = W(4) = \dots = W(8) = 1$ is supplied to MultiStart, which executes 100000 local solver runs starting in the set $[0,5]^6$. The output of MultiStart is then supplied to GlobalSearch as an initial guess, and the optimal value of the two solvers is taken as the best fit.

## Restricted models

To assess whether alternative fits weighted to later somite stages could explain the data as well as the best fit to the full model, four restricted models were also assessed. These models enforced in turn $W(3) = 0$, $W(3) = W(4) = 0$, $W(3) = W(4) = W(5) = 0$ and $W(3) = 0$.

## Acknowledgements

This work was mainly supported by the Fundação para a Ciência e Tecnologia (research grant – PTDC/BEX- BID/1411/2014).

PS was funded by a PhD fellowship FCT: SFRH/BD/111611/2015. SP was funded by a PhD fellowship FCT: SFRH/BD/130272/2017 CB was funded by a PhD fellowship FCT: SFRH/BD/141034/2018 IAT

was funded by FCT Investigator IF/00082/2013, EU FP7-PEOPLE-2013-CIG (N° 818743) and by the Fundação Calouste Gulbenkian (FCG). AG thanks to CZI Expanding Global Access to Bioimaging and to PAPIIT (IN211821) for funding support. DJS acknowledges support from the Engineering and Physical Sciences Research Council UK via a Healthcare Technologies Challenge Award (EP/N021096/1) and a Turing Fellowship (EP/N510129/1). SL was funded by FCT Investigator IF/00951/2012, by NOVA Medical School and by FCT CEEC-IND 2018. Zebrafish were reared and maintained in the NMS Fish Facility, with the support from the research consortia Congento funded by LISBOA-01–0145-FEDER-022170, microscopy was performed at the IGC Advanced Imaging Facility which is supported by PPBI-POCI-01–0145-FEDER-022122; all co-financed by Lisboa Regional Operational Program (Lisboa2020), under the Portugal 2020 Partnership Agreement, through the European Regional Development Fund (ERDF) and Fundação para a Ciência e Tecnologia under the project. The project LysoCil funded by the European Union Horizon 2020 research and innovation under grant agreement No 811087 supported SL, PS and SP travelling to the FASEB cilia meeting in the USA in 2019, where this work was presented and discussed with peers. We thank Sérgio Dias for reviewing the final version of the manuscript.

## Additional information

### Funding

| Funder | Grant reference number | Author |
|---|---|---|
| Fundação para a Ciência e a Tecnologia | FCT: SFRH/BD/130272/2017 | Sara Pestana |
| Fundação para a Ciência e a Tecnologia | PTDC/BEX- BID/1411/2014 | Susana Santos Lopes |
| Fundação para a Ciência e a Tecnologia | FCT: SFRH/BD/111611/2015 | Pedro Sampaio |
| Fundação para a Ciência e a Tecnologia | IF/00082/2013 | Ivo A Telley |
| Chan Zuckerberg Initiative LLC | Expanding Global Access to Bioimaging PAPIIT (IN211821) | Adán Guerrero |
| Fundação para a Ciência e a Tecnologia | FCT: SFRH/BD/141034/2018 | Catarina Bota |
| Alan Turing Institute | Turing Fellowship (EP/N510129/1) | David Smith |
| Engineering and Physical Sciences Research Council | Healthcare Technologies Challenge Award (EP/N021096/1) | David Smith |
| Fundação para a Ciência e a Tecnologia | FCT Investigator IF/00951/2012 | Susana Santos Lopes |
| NOVA Medical School | iNOVA4health unit | Susana Santos Lopes |
| Fundação para a Ciência e a Tecnologia | FCT CEEC-IND 2018. | Susana Santos Lopes |
| Fundação para a Ciência e a Tecnologia | EU FP7-PEOPLE-2013-CIG (N° 818743) | Ivo A Telley |

The funders had no role in study design, data collection and interpretation, or the decision to submit the work for publication.

### Author contributions

Pedro Sampaio, Formal analysis, Investigation, Methodology, Writing - original draft; Sara Pestana, Formal analysis, Investigation, Writing - original draft; Catarina Bota, Investigation, Methodology, Writing – review and editing; Adán Guerrero, Supervision, Writing – review and editing; Ivo A Telley,

Data curation, Formal analysis, Supervision, Methodology, Writing – review and editing; David Smith, Conceptualization, Data curation, Methodology, Writing – review and editing; Susana Santos Lopes, Conceptualization, Formal analysis, Supervision, Validation, Investigation, Visualization, Methodology, Project administration, Writing – review and editing

**Author ORCIDs**
Ivo A Telley ![ORCID] http://orcid.org/0000-0003-4444-1046
Susana Santos Lopes ![ORCID] http://orcid.org/0000-0002-6733-6356

**Decision letter and Author response**
Decision letter https://doi.org/10.7554/eLife.83861.sa1
Author response https://doi.org/10.7554/eLife.83861.sa2

## Additional files

**Supplementary files**
• Supplementary file 1. Motile and immotile cilia distribution at 5 somite stage embryos. Analyses of cilia motility per LRO anterior and posterior halves.
• MDAR checklist

### Data availability
All data generated or analysed during this study are included in the manuscript and supporting files.

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
