## [Editor Report]

Sampaio and colleagues manipulate fluid dynamics in zebrafish Kupffer's vesicle to ask if fluid movement or something in the fluid governs the break in symmetry. These results support a role for fluid movement and detection as important in breaking symmetry in a ciliated left-right organizer and help set a time window when fluid flow is critical for this process.

---

## [Decision Letter]

**Decision letter after peer review:**

Thank you for submitting your article "Fluid extraction from the left-right organizer uncovers mechanical properties needed for symmetry breaking" for consideration by *eLife*. Your article has been reviewed by 2 peer reviewers, and the evaluation has been overseen by a Reviewing Editor and Marianne Bronner as the Senior Editor. The reviewers have opted to remain anonymous.

Essential revisions:

1. Please revise the conclusions on the proposed mechanism and reduce the strong emphasis made on the role of mechanical forces. The set of experiments presented is not sufficient to make strong conclusions on the mechanism of the mechanosensory hypothesis because the fluid extraction experiments affect both chemical and physical features. This, it is not possible to disentangle the two with this approach. Please also attend to the comments of the reviewers as detailed below.

2. Please provide more context for the computational modeling.

3. It is important that the authors discuss their results in light of the recent papers of Hiroshi Hamada and Shiaolou Yuan published in Science. In particular, the Yuan lab shows that left-right symmetry breaking occurs between 1-4 somites in zebrafish which contrasts with what the Lopes lab proposes in their manuscript.

*Reviewer #1 (Recommendations for the authors):*

Here are some specific comments to help the authors to help improve their work for subsequent submission:

1) The authors claim that symmetry breaking occurs in one-hour intervals in the LRO, however, it looks like there is a trend for 3SS and 4SS embryos to present left-right defects, but the embryo number is extremely low for these datapoints (n=13 and n=26, respectively). Increasing the embryo number will certainly lead to statistical significance. The authors should provide dand5 expression after LRO liquid extraction in embryos at 3, 4, 5, 6, 8, and ss to validate their observation. The authors should provide alternative methods to conditional block flow should be provided to confirm this result (conditional inactivation of cilia motility) or other types of ablation.

2) About the mechanosensory hypothesis/chemosensory hypothesis. The authors try to show that the chemosensory hypothesis is not valid to demonstrate that it is the mechanosensory hypothesis that prevails. The authors exclude the chemical hypothesis based on the fluid replacement experiment, which is a negative result. The authors propose that since there are no left-right defects when we replace the KV fluids with inert fluids, this means that it cannot be a chemical factor at work in the system. However, the authors never exclude the possibility that once the fluids are replaced, the cells of the Kupffer's vesicle quickly secrete back any diffusible factor they can secrete. Without excluding this very likely possibility, this conclusion does not stand. Other important comments: Figure1c: the legend is the following: 'dand5 expression pattern after intervention for LRO liquid extraction in a subset of fixed embryos at 8 ss'. When was the intervention? How many embryos were analysed here?

Figure 3: why not show velocity? Why interval 4-6ss for LR establishment? The defective embryos show a reduced flow at 8ss; it is possible that the extraction at 5ss damages the flow and leads to defects during later readout? Figure2 and 3: Do the authors see a left-right difference in velocity? If not, they should comment on it.

Detailed comments on some conclusions:

"Indeed, if a signalling molecule contained in the LRO fluid was needed for dand5 mRNA degradation, then the prediction would be that the DB dilution experiment resulted in many more embryos with dand5 bilateral expression pattern than it did, as well as more heart misplacements." This conclusion is not valid. A signalling molecule that is present in the fluid over a longer time obviously can't discriminate between L and R. The chemosensitive hypothesis, as discussed in Ferreira 2017, proposes secretion in the anterior region and detection a few seconds later. Buffer replacement wouldn't affect this mechanism.

"Alternatively, if we attempt to interpret these results within a chemosensory mechanism, the dilution of a moving extracellular vesicle (EV) or molecule within could only occur without impacting LR if its secretion rate was sufficiently high to allow a fast turnover. This last hypothesis, however, is very unlikely and was refuted when we showed a limited response capacity to recover the asymmetric dand5 expression, after a second extraction intervention (Figure S4) that was performed one hour later than the first extractions in the same embryos (Figure 1c)." This part is very confusing: How can the second extraction experiment rule out the chemical hypothesis? Any chemical that can be secreted back within minutes in the LRO would explain these results very well.

"So, according to Ferreira et al. (2017), this fact brings to light that perhaps immotile cilia number is not a crucial factor at least in zebrafish." The authors should expand and explain better what they mean here. Reading Ferreira et al. in detail, I could not find any claim stating that the immotile cilia number is not a crucial factor.

"The fact that CBF is significantly lower in the methylcellulose injected embryos (Figure 4g) suggests that may be there is a minimum CBF threshold for the sensory system to generate efficient ICOs." The system would need to be very sensitive if a 25% frequency reduction messes up the mechanism. How about temperature fluctuation, etc.?

"However, our data does not show a consistent difference in left versus right angular velocity and correct LR development, leading us to suggest that anterior flow is the relevant factor not left." The authors should clarify how and what exactly is sensed by the system, is it flow direction or flow magnitude? They suggest that "Pkd1l1/ Pkd2 could be sensitive to flow magnitude" with no experiment to back this up. How flow magnitude could break left-right symmetry if not different between the right and left sides of the LRO?

"Our data highlights anterior flow magnitude as a major mechanical property in LR. Interventions that decreased flow magnitude or reduced its A/P difference were the ones that generated abnormal LR outcomes." How can this work? In any mechanosensitive scenario, the flow needs to be detected on the side. Anterior flow magnitude alone can't break the symmetry unless the authors propose a new mechanism.

Any methyl-cellulose injection: A mechanosensitive cilium can only detect forces, not velocities. So in a dense fluid, the forces should actually be enhanced. With chemicals, it is hard to say – the Peclet number will increase drastically, which improves the directionality but also reduces detection.

*Reviewer #2 (Recommendations for the authors):*

Suggestions to help improve the paper for readers:

1) The linear mixed-effects model isn't well explained and the figures that accompany this part of the article are hard to parse. Not being an expert in mathematical models, I am unable to get anything meaningful from these analyses. For example, I can't look at Table 1 or the explanation and understand what these numbers are supposed to indicate. I can't understand what the bars indicate in Figure 6 even though I understand the experiments that are supposedly represented. It would be helpful to have a bit more explanation for the audience for this information to be more impactful.

---

## [Author Response]

Essential revisions:1. Please revise the conclusions on the proposed mechanism and reduce the strong emphasis made on the role of mechanical forces. The set of experiments presented is not sufficient to make strong conclusions on the mechanism of the mechanosensory hypothesis because the fluid extraction experiments affect both chemical and physical features. This, it is not possible to disentangle the two with this approach. Please also attend to the comments of the reviewers as detailed below.

We have reduced the emphasis on mechanical by presenting always an alternative interpretation highlighting what fits or not the experimental data. Explanations to reviewers are showed below.

2. Please provide more context for the computational modeling.

We have made a great effort on this point as reflected by re-writing extensively the text (all text highlighted in yellow means new text).

3. It is important that the authors discuss their results in light of the recent papers of Hiroshi Hamada and Shiaolou Yuan published in Science. In particular, the Yuan lab shows that left-right symmetry breaking occurs between 1-4 somites in zebrafish which contrasts with what the Lopes lab proposes in their manuscript.

This point is now explained in page 24. Additionally, there may be a difference in the background of the zebrafish lines used between our lab and Yuan lab, as the KV lumen often opens at 2 ss (Oteíza et al., 2008). This point /possibility was also taken by Shiaolu Yuan and was later included in their published article.

Reviewer #1 (Recommendations for the authors):Here are some specific comments to help the authors to help improve their work for subsequent submission:1) The authors claim that symmetry breaking occurs in one-hour intervals in the LRO, however, it looks like there is a trend for 3SS and 4SS embryos to present left-right defects, but the embryo number is extremely low for these datapoints (n=13 and n=26, respectively). Increasing the embryo number will certainly lead to statistical significance.

One hour in zebrafish development means a period that is 2 somites apart (at 28ºC). So, we are considering 4 to 6 ss as the major interval for symmetry breaking, meaning that 4 ss is already included in the interval.

We do acknowledge the trend is there for 3 ss but experimentally, this is the most challenging stage as the KV is very, very small (lumen opens at 2 ss Oteíza et al., 2008). So, to perform extractions that still result in embryos that are able to refill the liquid (which was an important criteria) is not trivial and we tried it for longer that for the other stages. So, we have stopped trying at 13 embryos that show a trend and therefore we present a model that also shows 3 ss extractions are an important contributor for the symmetry breaking. So, we could indeed include 3 ss and state the relevant physiological interval is from 3 ss to 6 ss which translates in 1.5h. We thank the reviewer for pointing this out but statistically we need to keep the 4-6 ss window. We also note that the intervention disrupts flow at least half an hour (1 somite stage) and increases variability for much longer and so inferring critical period is not simple – which forms the motivation for building and fitting the probabilistic model of signal integration and symmetry breaking.

We also note that selectively conducting more experiments in pursuit of a statistical significance threshold constitutes ‘p-hacking’ and therefore would not be sound science at this stage.

The authors should provide dand5 expression after LRO liquid extraction in embryos at 3, 4, 5, 6, 8, and ss to validate their observation.

Regarding the request that *dand5* expression should be included after each extraction, we have to explain to the reviewer that such a request represents at least one year of work. I am afraid this is not possible to perform at this stage. Both Pedro and Sara are gone from the lab and I would have to train extensively to do it myself. Pedro trained one year to be good at this technique. Most important, our aim was indeed to look at organs as a readout because we wanted to find the most sensitive period of time that resulted in real *situs* defects (meaning organ defects) and also find the proportion of heterotaxy versus *situs inversus* cases. We did not want to use a readout that was early as 8 ss. We then explored in detail the extraction stage that showed more *situs* defects – 5 ss, at which we extracted more embryos and fixed embryos at 8 ss to perform *dand5 in situ* hybridizations. We had an extremely similar proportion of *dand5* mRNA defects compared to *situs* defects. So, we also think this request represents a duplication of work that will not add much to the manuscript.

The authors should provide alternative methods to conditional block flow should be provided to confirm this result (conditional inactivation of cilia motility) or other types of ablation.

It is known and it is extensively published that cilia immotility will result in LR defects. We did not want to show this. On the contrary, we wanted to keep every parameter intact as much as possible and only extract the fluid without affecting cilia motility. It is known that blocking cilia motility at 1 cell stage or by using cilia motility mutants, will generate random situs and provide stronger phenotypes than our extraction experiment. Alternatively, as suggested by the reviewer, conditional mutants for cilia immotility are not available and would represent another phD project at least. We would love to do it in the future with a new grant and new students but we are afraid this is impossible to perform at the moment.

2) About the mechanosensory hypothesis/chemosensory hypothesis. The authors try to show that the chemosensory hypothesis is not valid to demonstrate that it is the mechanosensory hypothesis that prevails. The authors exclude the chemical hypothesis based on the fluid replacement experiment, which is a negative result. The authors propose that since there are no left-right defects when we replace the KV fluids with inert fluids, this means that it cannot be a chemical factor at work in the system. However, the authors never exclude the possibility that once the fluids are replaced, the cells of the Kupffer's vesicle quickly secrete back any diffusible factor they can secrete. Without excluding this very likely possibility, this conclusion does not stand.

Thank you for pointing this out. We apologize because we had included a final paragraph in the discussion on this subject of the current manuscript that the reviewer missed entirely. Please read what we wrote regarding the potential secretion on the last page, last paragraph, last version. We have now expanded this section, but it was already included in the previous one.

Other important comments: Figure1c: the legend is the following: 'dand5 expression pattern after intervention for LRO liquid extraction in a subset of fixed embryos at 8 ss'. When was the intervention?

Thank you for pointing this out. The legend in Figure 1c says: *dand5* expression pattern at 8 ss after single and double intervention for LRO liquid extraction. The intervention was at 5 ss for the single extraction and at 5 and 7 ss for the double extractions. We added this information to the legend and Figure 1.

How many embryos were analysed here?

We are sorry, this information was only present in the text (in previous page 6) and we agree it should be copied to the figure legend in a more clear way (6/17 manipulated embryos at 5 ss showed *dand5* expression defects compared to 0/26 in non-manipulated controls, Fisher test, p-value = 0.002; Figure 1C) and 6/16 double extracted embryos at 5 and 7 ss showed dand5 expression defects. When comparing single and double extractions we found no significant difference (Fisher test p-value = 0.728).

Figure 3: why not show velocity?

We find this question confusing, velocity is a vectorial parameter, which velocity does the reviewer refer to? Angular velocity is shown and is the correct parameter to show in a spheroid organ. We have however, added in supplementary data, plots for tangential velocity and radial velocity (Supplemental Figures 4 and 5), which we mention on page 12 and discuss further now in page 27, 1^st^ paragraph.

Why interval 4-6ss for LR establishment?

Because our experimental data in Figure 1 shows this is the physiological interval at which embryos are more sensitive to the fluid extraction manipulation. The probabilistic model of signal integration also shows that only models in which there is sensitivity to flow between 4-6ss are capable of making a satisfactory fit to experimental data (compare Figures 6A, D, G, J and M).

The defective embryos show a reduced flow at 8ss; it is possible that the extraction at 5ss damages the flow and leads to defects during later readout?

At 8 ss angular velocity is indeed reduced. Yes, it is possible. We show that the embryos that show defects have lower anterior angular velocity patterns and have slower recovery over time (Table 1 shows all the significant parameters that are changed).

Figure2 and 3: Do the authors see a left-right difference in velocity? If not, they should comment on it

We did observe a significant LR difference in angular velocity. This is clearly shown in the p-values in Table 1. From Table 1 we can extract that for the no-defects group we saw a LR axis difference (p=0.006176 **) for the defects group we lose this difference (p=0.79567) for the sham controls we did not see a LR difference (p=0.57655).

The multivariable linear regression of Table 1 shows that there is no significant trend in left-right position affecting angular velocity, but there is a trend within the ‘no-defects’ group. This is commented on page 10 ‘(iv) enhanced left-right difference in the ‘No Defects group’ (p-value=0.006, coefficient 0.027 rad/s)’.

.Detailed comments on some conclusions:"Indeed, if a signalling molecule contained in the LRO fluid was needed for dand5 mRNA degradation, then the prediction would be that the DB dilution experiment resulted in many more embryos with dand5 bilateral expression pattern than it did, as well as more heart misplacements." This conclusion is not valid. A signalling molecule that is present in the fluid over a longer time obviously can't discriminate between L and R. The chemosensitive hypothesis, as discussed in Ferreira 2017, proposes secretion in the anterior region and detection a few seconds later. Buffer replacement wouldn't affect this mechanism.

Here we disagree with the reviewer. Buffer replacement affects this mechanism at a short time scale. Again, the predictions by Ferreira et al. (2017) are just predictions. Nobody measured any signaling molecule dynamics yet. Here we performed direct manipulations to the fluid that must result in mixing and advection at a short scale and are denoted in our results at Table 2 when looking at Interaction: DB injection and time (p=0.0099937 ***).

"Alternatively, if we attempt to interpret these results within a chemosensory mechanism, the dilution of a moving extracellular vesicle (EV) or molecule within could only occur without impacting LR if its secretion rate was sufficiently high to allow a fast turnover. This last hypothesis, however, is very unlikely and was refuted when we showed a limited response capacity to recover the asymmetric dand5 expression, after a second extraction intervention (Figure S4) that was performed one hour later than the first extractions in the same embryos (Figure 1c)." This part is very confusing: How can the second extraction experiment rule out the chemical hypothesis? Any chemical that can be secreted back within minutes in the LRO would explain these results very well.

Yes, exactly! but it did not happen that way. In the double extraction experiment we did not have recovery of *dand5* expression, which suggests that any potential secretion of molecules did not occur at a sufficiently fast rate. So, in the end, *dand5* defects were maintained in single and double extractions (Figure 1). We alert and inform the reviewer that fixation for *dand5* in situs occurs at 8 ss, which is 2 hours later. However, no recovery of *dand5* expression was noted, even after 2 hours of potential secretion. This is now stated in the end of the Discussion section because it is an important fact to correctly interpret the replacement experiment.

"So, according to Ferreira et al. (2017), this fact brings to light that perhaps immotile cilia number is not a crucial factor at least in zebrafish." The authors should expand and explain better what they mean here. Reading Ferreira et al. in detail, I could not find any claim stating that the immotile cilia number is not a crucial factor.

Thank you for pointing that this point needs further explanation. In the abstract of Ferreira et al. (2017) DOI: 10.7554/*eLife*.25078, one can read:

“Our results show that the number of immotile cilia is too small to ensure robust left and right determination by mechanosensing, given the large spatial variability of the flow. However, motile cilia could sense their own motion by a yet unknown mechanism”.

This conclusion is given in the abstract because the modelling performed therein led to that statement. In our own argumentation we are considering this conclusion by the Vermot lab and comparing it to our experimental data on supplementary Table 1 and agreeing that they may be correct given that immotile cilia number was also low in our hands, in this present experiment.

"The fact that CBF is significantly lower in the methylcellulose injected embryos (Figure 4g) suggests that may be there is a minimum CBF threshold for the sensory system to generate efficient ICOs." The system would need to be very sensitive if a 25% frequency reduction messes up the mechanism. How about temperature fluctuation, etc.?

Room temperature was controlled throughout these experiments. I want to bring to light that in Yuan lab last publication (Djenoune et al., 2023) they also noted that the system is very sensitive, and even suggest that noise may be filtered at different levels in WT versus no-flow mutants:

“We posit that the cilium’s noise filtering capability may be adaptable, as the filter threshold in WT embryos with intrinsic flow appears to be higher than in flowless embryos as suggested by the lower number of responses in WT LRO cilia (Figure 3, I and J) compared with c21orf59 cilia (Figure 3, C and D).”

"However, our data does not show a consistent difference in left versus right angular velocity and correct LR development, leading us to suggest that anterior flow is the relevant factor not left." The authors should clarify how and what exactly is sensed by the system, is it flow direction or flow magnitude? They suggest that "Pkd1l1/ Pkd2 could be sensitive to flow magnitude" with no experiment to back this up. How flow magnitude could break left-right symmetry if not different between the right and left sides of the LRO?

We thank the reviewer for spotting this confusing point. We have changed the word magnitude to sustained flow angular velocity (page 26), which is what we tested. We, of course do not know the correct mechanism, as nobody else knows. We were speculating that as Pkd1l1 and Pkd2 are present in all cilia of the zebrafish LRO (see our immunostainings in Roxo-Rosa & Lopes, 2017; Jacinto et al. 2019) a directional sensing capacity (as now observed in mouse node by Hamada’s lab) seemed impossible and therefore we hypothesized that what was being sensed was velocity magnitude = speed of flow, because in our hands and also in Amack’s lab previous work, keeping the ratio between anterior and posterior number of cilia or flow angular velocity seems to matter more. However, recent data from Hamada´s lab (Katoh et al. 2023) has shed light into this problem at least for the mouse system. We discuss this point extensively throughout the manuscript.

Regarding left and right: setting a clear difference anterior-posteriorly immediately generates a difference L and R because of the inherent counterclockwise direction of flow. So, anterior higher angular velocity causes a right to left flow anteriorly that is stronger than a left to right flow posteriorly. This is stated in the manuscript in the results and Discussion sections.

"Our data highlights anterior flow magnitude as a major mechanical property in LR. Interventions that decreased flow magnitude or reduced its A/P difference were the ones that generated abnormal LR outcomes." How can this work? In any mechanosensitive scenario, the flow needs to be detected on the side. Anterior flow magnitude alone can't break the symmetry unless the authors propose a new mechanism.

Anterior whirling flow is directed towards the left side of KV, so if the anterior flow were dominant, then a mechanosensory mechanism would only need to be able to differentiate flow incoming to the left compared with outgoing from the right. The data of Katoh et al. (Science 2023) suggest that differentiation of incoming versus outgoing flow could be achieved in the mouse node through asymmetric location of Pkd2. So, we don’t believe a new mechanism is necessary here, rather a translation between the whirling flow in KV and leftward flow in mouse node.

Any methyl-cellulose injection: A mechanosensitive cilium can only detect forces, not velocities. So in a dense fluid, the forces should actually be enhanced. With chemicals, it is hard to say – the Peclet number will increase drastically, which improves the directionality but also reduces detection.

We interpret the reviewer as meaning ‘viscous’ rather than ‘dense’ here (as MC minimally affects density). An increase in viscosity will increase forces only if the flow magnitude is the same. However, we know that increasing viscosity significantly affects cilia beat frequency and amplitude, and hence fluid velocity (Table 2 line 3). What is more, MC injection also appears to reduce anterior-posterior asymmetry, which is key to establishing situs. We are unsure of the precise mechanism of how MC reduces AP asymmetry but it is clear that once this effect has occurred, signal transduction will not be enhanced.

Reviewer #2 (Recommendations for the authors):Suggestions to help improve the paper for readers:1) The linear mixed-effects model isn't well explained and the figures that accompany this part of the article are hard to parse. Not being an expert in mathematical models, I am unable to get anything meaningful from these analyses. For example, I can't look at Table 1 or the explanation and understand what these numbers are supposed to indicate. I can't understand what the bars indicate in Figure 6 even though I understand the experiments that are supposedly represented. It would be helpful to have a bit more explanation for the audience for this information to be more impactful.

The comment conflatestwo distinct quantitative analyses, which we acknowledge indicates that greater clarity is needed in the manuscript. We also acknowledge that we should have included more discussion of the findings.

The first quantitative analysis is the linear mixed effects regression analysis of each experiment (in our revision we now replace ‘model’ with ‘regression’ to minimise confusion) which seeks trends in the angular velocity data depending on location in the node, and taking into account somite stage and group (in Table 1 this is sham/no defects/defects and in Table 2 this is sham/DB dilution/MC dilution). We are looking for trends here rather than a simple group versus group analysis. Mixed effects regression takes account of the issue that repeated observations from the same embryo are not independent. (There is a good introduction to linear mixed effects models aimed at a general audience here: https://www.frontiersin.org/articles/10.3389/fpsyg.2015.00002/full)

We did try to interpret results better as we indicate them and included a summary/interpretation of the findings of the linear mixed effects regression of Table 1 in the Discussion section: see all highlighted text please.

‘In embryos with LR defects we found a general decreased anterior angular velocity, a smaller difference between anterior-posterior angular velocity and an abnormal flow directionality in anterior sections of the LRO.’

We have also expanded on the discussion session to integrate a summary of dilution and viscosity experiments and analysis of Table 2:

We showed that if the fluid extraction is made until 6 ss the system senses it and there is a response that may compensate the insult but if the extraction is made at 7 ss the LRO does not sense it anymore. This heightened the need of identifying the mechanism that senses the challenge in the relevant short time-interval.

We therefore assessed the effect of replacing extracted fluid with either buffer, which simply dilutes the KV contents, or buffer with added methylcellulose, which also increases viscosity. Simple dilution did not significantly affect situs whereas increasing viscosity dramatically increased rates of right situs. Statistical analysis of flow data showed that increasing the viscosity via injection is associated with very strong reduction in angular velocity, a strong reduction in anterior-dominance of the flow, and modestly faster flow on the right compared with the left. By contrast, dilution which does not increase viscosity is associated with a modest reduction in anterior-dominance and a modest reduction in flow acceleration over time. The strong effects appear to explain the effect of viscosity on situs. We therefore concluded that the sensory mechanism detects flow properties, as angular velocity and viscosity but seems to be insensitive if we quickly replace the fluid and keep flow dynamics constant.

The second type of quantitative analysis is the probabilistic mathematical model of signal sensing and integration which is used to reinforce the time window analysis, in addition to presenting a conceptual framework for how flow perturbations influence situs (including by increasing the variance of flow velocity as well as a transient reduction). The model parameters to be determined by fitting to experimental data are the sensitivity weightings of each stage. The model fit is what is shown in Figure 6. The figure caption refers to ‘stage weightings’ and ‘stage contributions’, which quantify how important each stage is in breaking situs (the first being a sensitivity, the latter taking account of flow magnitude also). A tall bar corresponds to that stage being important. While a description of the model was given in the main text, we have revised the text to improve accessibility. The relevant part of the Results section is revised as follows:

“Taking account both stage weighting and anterior angular velocity to quantify the contribution of each stage, nearly 90% of the contribution to symmetry breaking occurred in the 4 ss – 6 ss interval (Figure 6C; note that tall bars indicate a high contribution relative to other stages).”

We have also rewritten the Figure 6 caption to guide the reader through the figure:

“Figure 6 – Mathematical modelling confirms the critical role of one-hour interval confirming the experimental observations. Outcome of model fitting to experiment, associated stage weightings (sensitivity of KV to its own flow) and signal contribution (sensitivity multiplied by flow strength) for (A-C) Unrestricted model, (D-F) model with 3 ss contribution omitted, (G-I) model with 3-4 ss contributions omitted, (J-L) model with 3-5 ss contributions omitted, (M-O) model with 3-6 ss contributions omitted. The restricted models are included to assess alternative explanations for the data in which later stages still make a significant contribution. (A, D, G, J, M) fitted model outputs of rates of normal and abnormal situs for direct comparison with experiment; (B, E, H, K, N) fitted stage weightings (sensitivity of KV to its own flow) where tall bars mean ‘relatively more sensitive’ in arbitrary units, (C, F, I, L, O) signal contribution (sensitivity multiplied by flow strength) where tall bars mean ‘relatively more contribution’. The fitted restricted models for (G-I, J-L and M-O) do not fit the experiment well which confirms that they are unlikely to represent what is occurring in the KV. The fitted models for (A-C, D-F and G-I) have a much better fit and confirm that 5ss makes a major contribution, as do 4ss and/or 6ss. ss: somite stage.”

Finally we have modified the relevant paragraph of the Discussion:

“After fluid extraction, the time lag and complexity of the fluid flow recovery process motivated the development of a mathematical model of flow, its variability, and the effect of experimental interventions, with the aim of determining the sensitivity of KV to its own flow field through fitting experimental data. The core idea of the model is to represent symmetry breaking as a random process which is biased by flow. The analysis took account of natural variability in the flow field and its modification by intervention, with the biasing effects of each stage to be determined through fitting, and for a comparison to be made against alternative models in which only later developmental stages are important. This analysis confirmed the critical importance of 4 ss to 6 ss in symmetry breaking, and the limited role for flow during 7 ss and 8 ss.”